# Finetuning-free Alignment of Diffusion Model for Text-to-Image Generation

## Abstract

Diffusion models have demonstrated remarkable success in text-to-image generation. While many existing alignment methods primarily focus on fine-tuning pre-trained diffusion models to maximize a given reward function, these approaches require extensive computational resources and may not generalize well across different objectives. In this work, we propose a novel finetuning-free alignment framework by leveraging the underlying nature of the alignment problem—sampling from reward-weighted distributions. Moreover, we give an in-depth discussion of adopting current guidance methods for text-to-image alignment. We identify a fundamental challenge: the adversarial nature of the guidance term can introduce undesirable artifacts in the generated images. To address this, we propose a regularization strategy that stabilizes the guidance signal. We evaluate our approach on a text-to-image benchmark and demonstrate comparable performance to state-of-the-art models with one-step generation while achieving at least a 60% reduction in computational cost.

## 1 Introduction

Diffusion models have achieved impressive performance in text-to-image generation, as demonstrated by state-of-the-art models such as Imagen (Saharia et al., 2022), DALL-E 3 (Betker et al.), and Stable Diffusion (Rombach et al., 2021). These models have been proven capable of generating high-quality, creative images even from novel and complex text prompts.

Inspired by Reinforcement Learning from Human Feedback (RLHF) (Ouyang et al., 2022), many alignment approaches leverage preference pairs to fine-tune models for generating samples that align with task-specific objectives. RLHF-type methods (Lee et al., 2023; Fan et al., 2023; Black et al., 2023; Clark et al., 2024; Chakraborty et al., 2024) typically learn a reward function and then use the policy gradients (Jaques et al., 2016; 2020) to update the model. On the other hand, Direct Preference Optimization (DPO)-type methods (Rafailov et al., 2024; Wallace et al., 2023; Yang et al., 2023; Liang et al., 2024; Yang et al., 2024) directly optimize the model to adhere to human preferences, without requiring explicit reward modeling or reinforcement learning.

Despite their effectiveness, these approaches require modifying model parameters through fine-tuning, which comes with several limitations. For example, fine-tuning for new reward functions is computationally expensive and often requires carefully designed training strategies; otherwise, optimizing on a limited set of input prompts can limit generalization to unseen prompts. More importantly, existing fine-tuning approaches do not fully exploit the structure of the alignment problem. Instead, they typically apply Low-Rank Adaptation (LoRA) to optimize model weights for a specific reward function, which may not be the most efficient strategy.

In contrast, plug-and-play alignment methods integrate new objectives without modifying the underlying model parameters, significantly reducing computational costs while adapting flexibly to different reward functions. In this paper, we develop a plug-and-play guidance term for the text-to-image diffusion models. Instead of treating alignment purely as a fine-tuning problem, we formulate it as a sampling problem from a reward-weighted distribution, leveraging its unique structure. We demonstrate that the score function required for this reward-weighted distribution can be effectively decomposed into the pre-trained score function with an additional guidance term. However, we also identify a critical issue: it is hard to determine the strength of the guidance, stemming from the adversarial nature of the guidance. To address this, we introduce a novel regularization term that

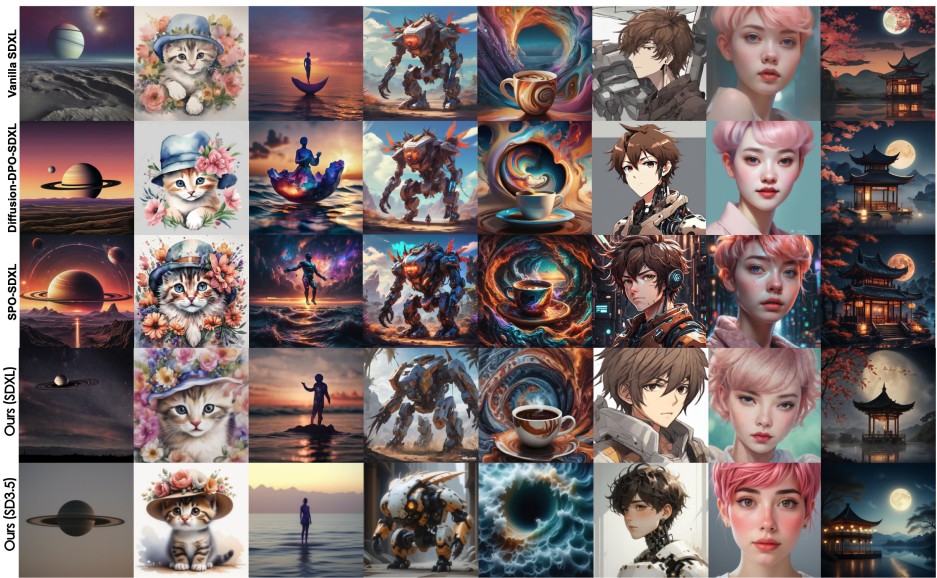

Figure 1: Qualitative comparison with Vanilla SDXL, Diffusion-DPO, and SPO. Our method achieves better aesthetic quality and stronger alignment with the text prompt. Prompts are provided in the Appendix B.3.

mitigates this issue and empirically validates its effectiveness on a text-to-image benchmark with one-step generation.

Our contributions can be summarized as follows.

- We identify that the alignment problem exhibits a particular structure well-suited for guidance-based methods—namely, alignment can be framed as sampling from a reward-weighted distribution. Based this insight, we introduce a finetuning-free alignment framework that leverages guidance to achieve alignment efficiently.

- We uncover a critical challenge in applying guidance-based methods to text-to-image diffusion models: the adversarial nature of guidance can lead to undesirable artifacts in the generated images, compromising visual quality and alignment with human preferences.

- We propose a regularization technique for training the guidance network, mitigating the identified issue and achieving strong performance on text-to-image alignment benchmarks. Furthermore, we demonstrate the effectiveness of our approach in the one-step generation setting, significantly reducing computational costs.

## 1.1 RELATED WORK

Existing alignment methods can be broadly categorized into two approaches: RLHF-based method that uses policy gradient to update the diffusion models, and DPO-based methods that use a parametrization trick to update the diffusion models without explicitly learning the reward function.

**RLHF-based alignment of diffusion model.** The authors first train a reward model to predict human feedback and adopt a reward-weighted finetuning objective to align the diffusion model in (Lee et al., 2023). In (Fan et al., 2023; Black et al., 2023), the authors use policy gradient algorithms to update the diffusion models under Kullback–Leibler (KL) constraints. The authors propagate the reward function gradient through the full sampling procedure in (Clark et al., 2024). They reduce the memory costs by adopting low-rank adaptation (LoRA) (Hu et al., 2021) and gradient checkpointing (Chen et al., 2016).

**DPO-based alignment of diffusion model.** This line of works (Wallace et al., 2023; Yang et al., 2023) directly applies DPO (Rafailov et al., 2024) to align the diffusion model with human preference. In (Liang et al., 2024), Liang et al. propose a step-aware preference model and a step-wise resampler to align the preference optimization target with the denoising performance at each timestep. The authors take on a finer dense reward perspective and derive a tractable alignment objective that emphasizes the initial steps in (Yang et al., 2024).

**Training-free guidance.** This line of work (Chung et al., 2022; Graikos et al., 2022; Lu et al., 2023; Song et al., 2023; Bansal et al., 2023; Yu et al., 2023; Shen et al., 2024; Ye et al., 2024) explores the use of diffusion models as plug-and-play priors for solving inverse problems. Recently, several studies (Li et al., 2024) have focused on adopting training-free guidance to enhance alignment in language tasks, particularly in discrete diffusion models. Some work Shen et al. (2024); Tang et al. (2024); Uehara et al. (2024); Ma et al. (2025); Singhal et al. (2025) study inference-time optimization for alignment. However, to the best of our knowledge, there has been limited exploration of applying guidance to address the challenge of text-to-image alignment in the context of one-step generation. This gap motivates our work.

## 2 PRELIMINARY

In this section, we focus on existing techniques for aligning pre-trained models with human preferences. We first provide a brief description of Diffusion Model in Section 2.1. Then, we decompose the alignment procedure into two important components, reward learning to model the human preference in Section 2.2 and the alignment methods in Section 2.3.

### 2.1 DIFFUSION MODELS

The diffusion model (Ho et al., 2020; Song et al., 2021) first gradually injects Gaussian noise into samples $\mathbf{x}_0$ from the data distribution by following the stochastic differential equation:

$$\mathrm{d}\mathbf{x}_t = \mathbf{f}(\mathbf{x}_t, t)\mathrm{d}t + g(t)\mathrm{d}\mathbf{w}, \ t \in [0, T], \tag{1}$$

where $\mathbf{w}$ is the standard Brownian motion, $\mathbf{f}(\cdot, t) : \mathbb{R}^d \to \mathbb{R}^d$ is a drift coefficient, and $g(\cdot) : \mathbb{R} \to \mathbb{R}$ is a diffusion coefficient. We use $p_t(\mathbf{x})$ to denote the marginal distribution of $\mathbf{x}_t$ at time $t$. And we can use the time reversal of equation 1 for generation, which admits the following form (Anderson, 1982):

$$\mathrm{d}\mathbf{x}_t = \left[\mathbf{f}(\mathbf{x}_t, t) - g(t)^2 \nabla_{\mathbf{x}} \log p_t(\mathbf{x})\right] \mathrm{d}t + g(t)\mathrm{d}\overline{\mathbf{w}}, \tag{2}$$

where $\overline{\mathbf{w}}$ is a standard Brownian motion when time flows backwards from $T$ to 0, and $\mathrm{d}t$ is an infinitesimal negative time step. The score function of each marginal distribution $\nabla_{\mathbf{x}} \log p_t(\mathbf{x})$ needs to be estimated by the following score matching objective:

$$\min_{\boldsymbol{\theta}} \mathbb{E}_t \left\{ \lambda(t) \mathbb{E}_{p_t(\mathbf{x}_t)} \left[ \|\mathbf{s}_{\boldsymbol{\theta}}(\mathbf{x}_t, t) - \nabla_{\mathbf{x}_t} \log p_t(\mathbf{x}_t)\|_2^2 \right] \right\}, \tag{3}$$

where $\lambda(t) : [0, T] \to \mathbb{R}_{>0}$ is a positive weighting function, $t$ is uniformly sampled over $[0, T]$. The latent diffusion model (Rombach et al., 2021; Podell et al., 2023) further extends diffusion models to text-to-image generation. They use an image encoder $\mathcal{E}$ that maps $\mathbf{x}$ into a latent representation and use a text encoder $\tau$ that maps the prompts $y$ into an embedding as the condition.

### 2.2 REWARD LEARNING

The Bradley-Terry (BT) models (Bradley & Terry, 1952) (or more general Plackett-Luce ranking models (Plackett, 1975; Luce, 1979)) are a popular way to model preferences. Given a prompt $y$ and a pair of answers $\mathbf{x}_w \succ \mathbf{x}_l \mid y$, where $\mathbf{x}_w$ denotes the winning response and $\mathbf{x}_l$ denotes the losing response under the preference of humans. The BT model depicts the preference distribution as

$$p(\mathbf{x}_w \succ \mathbf{x}_l \mid y) = \frac{\exp(r(\mathbf{x}_w, y))}{\exp(r(\mathbf{x}_w, y)) + \exp(r(\mathbf{x}_l, y))},$$

where $r(\mathbf{x}, y)$ denotes the reward model and can be learned by the following maximum likelihood objective,

$$\min_{\boldsymbol{\phi}} -\mathbb{E}_{(\mathbf{x}_w, \mathbf{x}_l, y) \sim \mathcal{D}} \left[ \log \sigma(r(\mathbf{x}_w, y) - r(\mathbf{x}_l, y)) \right], \tag{4}$$

where $\mathcal{D} = \left\{ \mathbf{x}_w^{(i)}, \mathbf{x}_l^{(i)}, y^{(i)} \right\}_{i=1}^N$ is the offline preference dataset and $\sigma$ denotes the logistic function.

## 2.3 ALIGNMENT OF DIFFUSION MODEL

Building on the success of alignment techniques for finetuning large pre-trained language models, many studies have explored aligning diffusion models with human preferences. A detailed discussion follows.

**RLHF.** This type of works (Lee et al., 2023; Xu et al., 2023; Fan et al., 2023; Black et al., 2023; Clark et al., 2024) finetune the pre-trained model $\pi_{\text{ref}}$ by policy gradient objective (Jaques et al., 2016; 2020),

$$\max_{\pi_\theta} \mathbb{E}_{y \sim \mathcal{D}_{\text{prompt}}, \mathbf{x} \sim \pi_\theta(\mathbf{x}|y)} \left[ r(\mathbf{x}, y) \right] - \beta \mathbb{D}_{\text{KL}} \left[ \pi_\theta(\mathbf{x} \mid y) \| \pi_{\text{ref}}(\mathbf{x} \mid y) \right], \tag{5}$$

where $\mathcal{D}_{\text{prompt}}$ denotes the prompt dataset. This type of method requires a pre-trained reward function for policy optimization (Schulman et al., 2017).

**DPO.** The authors propose not to explicitly learn the reward function in (Rafailov et al., 2024). They start with the analytic solution of equation 5 as the energy-guided form,

$$\pi_{\text{r}}(\mathbf{x} \mid y) = \frac{1}{Z(y)} \pi_{\text{ref}}(\mathbf{x} \mid y) \exp \left( \frac{1}{\beta} r(\mathbf{x}, y) \right), \tag{6}$$

where $Z(y) = \int \pi_{\text{ref}}(\mathbf{x} \mid y) \exp \left( \frac{1}{\beta} r(\mathbf{x}, y) \right) \mathrm{d}\mathbf{x}$ is the partition function. Therefore, they can reparametrize the reward function $r(\mathbf{x}, y)$ as

$$r(\mathbf{x}, y) = \beta \log \frac{\pi_{\text{r}}(\mathbf{x} \mid y)}{\pi_{\text{ref}}(\mathbf{x} \mid y)} + \beta \log Z(y). \tag{7}$$

Plugging equation 7 into equation 4, we yield the objective of DPO-type methods:

$$\min -\mathbb{E}_{(\mathbf{x}_w, \mathbf{x}_l, y) \sim \mathcal{D}} \left[ \log \sigma \left( \beta \log \frac{\pi_\theta(\mathbf{x}_w \mid y)}{\pi_{\text{ref}}(\mathbf{x}_w \mid y)} - \beta \log \frac{\pi_\theta(\mathbf{x}_l \mid y)}{\pi_{\text{ref}}(\mathbf{x}_l \mid y)} \right) \right]. \tag{8}$$

## 3 METHOD

In this section, we introduce the proposed finetuning free method to directly sample from the reward-guided distribution. We introduce the methodology formulation in Section 3.1. We provide an in-depth analysis of several vanilla methods for calculating the guidance, as discussed in Section 3.2. We highlight that these vanilla guidance methods exhibit adversarial guidance, which generates undesirable artifacts and worsens performance, particularly in text-to-image generation. Then, we present an enhanced method in Section 3.3 that alleviates the problem.

## 3.1 METHODOLOGY FORMULATION

Inspired by previous works from transfer learning (Ouyang et al., 2024), we consider preference learning in terms of transferring a pre-trained diffusion model to adapt to the given preference data. To this end, we propose a finetuning-free alignment method for the diffusion models. Instead of using RLHF-type (like equation 5) or DPO-type (like equation 8) alignments, we propose to directly sample from the reward-weighted distribution $\pi_{\text{r}}(\mathbf{x} \mid y)$ in equation 6 leveraging the relationships between score functions in the following Theorem.

**Theorem 3.1.** *Let the conditional distribution of reference diffusion model $\pi_{ref}(\mathbf{x}|y)$ be denoted as distribution $p$ and the reward-weighted distribution $\pi_r(\mathbf{x}|y)$ defined in equation 6 as distribution $q$. Under some mild assumption of the forward noising process detailed in Appendix A, let $\phi^*$ be the optimal solution for the conditional diffusion model trained on target domain $q(\mathbf{x}_0, y)$, i.e.,*

$$\phi^* = \arg\min_{\phi} \mathbb{E}_t \left\{ \lambda(t) \mathbb{E}_{q_t(\mathbf{x}_t, y)} \left[ \left\| \mathbf{s}_\phi(\mathbf{x}_t, y, t) - \nabla_{\mathbf{x}_t} \log q_t(\mathbf{x}_t|y) \right\|_2^2 \right] \right\}, \tag{9}$$

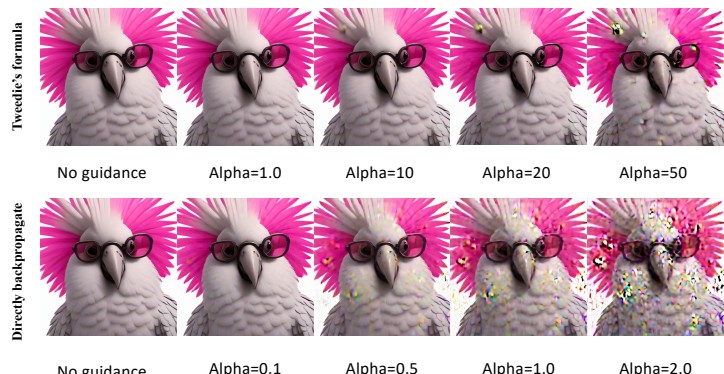

Figure 2: Illustration of the Adversarial Nature of Guidance. When the strength of the guidance is too small, there is little difference between the generated images with or without guidance. However, as the magnitude of the guidance increases (from left to right), undesirable artifacts become more pronounced. The prompt is "A 3D Rendering of a cockatoo wearing sunglasses. The sunglasses have a deep black frame with bright pink lenses. Fashion photography, volumetric lighting, CG rendering".

*then*

$$\mathbf{s}_{\phi^*}(\mathbf{x}_t, y, t) = \underbrace{\nabla_{\mathbf{x}_t} \log p_t(\mathbf{x}_t|y)}_{\substack{\text{pre-trained conditional model} \\ \text{on source}}} + \underbrace{\nabla_{\mathbf{x}_t} \log \mathbb{E}_{p(\mathbf{x}_0|\mathbf{x}_t, y)} \left[ \exp(\frac{1}{\beta} r(\mathbf{x}_0, y)) \right]}_{\text{conditional guidance}}. \qquad (10)$$

The proof can be found in Appendix A. Based on equation 19, we can calculate the additional guidance term rather than finetuning the text-to-image generative model by RLHF-type (like equation 5) or DPO-type (like equation 8). In general, the guidance term in equation 19 is not easy to compute because we need to sample from $p(\mathbf{x}_0|\mathbf{x}_t, y)$ for each $\mathbf{x}_t$ in the generation process. In the following, we first discussion the existing ways to calculate the guidance term.

### 3.2 VANILLA METHOD TO COMPUTE THE GUIDANCE TERM

**M1: Direct backpropagate through diffusion process.** The first method directly backpropagates through diffusion process to calculate $\nabla_{\mathbf{x}_t} \log \mathbb{E}_{p(\mathbf{x}_0|\mathbf{x}_t, y)} \left[ \exp(\frac{1}{\beta} r(\mathbf{x}_0, y)) \right]$ for fine-tuning the diffusion model. In Song et al. (2023), the author proposes an unbiased Monte Carlo estimation:

$$\nabla_{\mathbf{x}_t} \log \mathbb{E}_{p(\mathbf{x}_0|\mathbf{x}_t, y)} \left[ \exp \left( \frac{1}{\beta} r(\mathbf{x}_0, y) \right) \right] \approx \nabla_{\mathbf{x}_t} \log \frac{1}{n} \sum_{i=1}^{n} \exp \left( \frac{1}{\beta} r(\mathbf{x}_0^i, y) \right), \qquad (11)$$

where $\mathbf{x}_0^i$ denotes the $i$-th sample drawn from $p(\mathbf{x}_0|\mathbf{x}_t, y)$. However, this Monte Carlo estimation significantly increases memory costs, especially in text-to-image generation. Inspired by recent studies (Clark et al., 2024), we can borrow the same techniques, e.g., accumulated gradients along the diffusion process using techniques such as low-rank adaptation (LoRA) (Hu et al., 2021) and truncation or gradient checkpointing (Prabhudesai et al., 2023; Clark et al., 2024), to alleviate the memory cost of backpropagate through the diffusion process for calculating the guidance term. We can further reduce the memory cost by using the few-step diffusion model as the reference model. Despite these techniques, the memory requirements remain higher compared to the proposed approach.

**M2: Approximate and apply Tweedie's formula.** The second method first approximates the guidance term inspired by (Chung et al., 2022):

$$\nabla_{\mathbf{x}_t} \log \mathbb{E}_{p(\mathbf{x}_0|\mathbf{x}_t,y)} \big[ \exp\big(\frac{1}{\beta} r(\mathbf{x}_0, y)\big) \big] \approx \frac{1}{\beta} \nabla_{\mathbf{x}_t} r\big(\mathbb{E}_{p(\mathbf{x}_0|\mathbf{x}_t,y)}[\mathbf{x}_0], y\big). \tag{12}$$

Then, Tweedie's formula is further applied by Bansal et al. (2023); Chung et al. (2022); Yu et al. (2023):

$$\mathbb{E}\left[\boldsymbol{x}_0 \mid \boldsymbol{x}_t, y\right] = \boldsymbol{x}_t + \sigma_t^2 \nabla_{\boldsymbol{x}_t} \log p_t\left(\boldsymbol{x}_t|y\right).$$

However, as noted by Lu et al. (2023); Song et al. (2023), the approximation used in equation 12 is biased, leading to an incorrect calculation of the guidance term.

In the following, we empirically evaluate the effectiveness of these methods for aligning text-to-image generation tasks. We first identify a previously overlooked issue that contributes to suboptimal alignment performance. Fig 2 illustrates the performance of two vanilla methods under the guidance of PickScore (Kirstain et al., 2023), a reward function that evaluates whether the generated images align with human aesthetic and semantic preferences. The x-axis represents the strength of the guidance term, denoted by $\alpha$ [1]. Our experiments reveal that tuning this hyperparameter presents significant challenges. Insufficient values of $\alpha$ produce results indistinguishable from unguided generation, while excessive values introduce substantial artifacts that degrade image quality.

We attribute this phenomenon to the adversarial nature of the guidance mechanism, as observed in prior work (Shen et al., 2024). In equation 19, the guidance term is directly added to the estimated score. If the landscape is not smooth or does not behave well [2], the adversarial nature of the guidance can lead to undesirable artifacts in the generated images. To address these limitations, our proposed framework provides theoretical guarantees for generating properly aligned distributions with a fixed strength parameter $\alpha = 1$. Furthermore, we develop an additional regularization technique for training the guidance network that mitigates these instability issues.

### 3.3 PROPOSED FINETUNING-FREE GUIDANCE

We first utilize the following trick to calculate the conditional expectation. This trick has been used in previous works such as (Ouyang et al., 2024; Lu et al., 2023).

**Lemma 3.2.** *For a neural network $h_{\boldsymbol{\psi}}(\mathbf{x}_t, y, t)$ parameterized by $\boldsymbol{\psi}$, define the objective*

$$\mathcal{L}_{guidance}(\boldsymbol{\psi}) := \mathbb{E}_{p(\mathbf{x}_0, \mathbf{x}_t, y)} \left[ \left\| h_{\boldsymbol{\psi}}(\mathbf{x}_t, y, t) - \exp(\frac{1}{\beta} r(\mathbf{x}_0, y)) \right\|_2^2 \right], \tag{13}$$

*then its minimizer $\boldsymbol{\psi}^* = \arg\min_{\boldsymbol{\psi}} \mathcal{L}_{guidance}(\boldsymbol{\psi})$ satisfies:*

$$h_{\boldsymbol{\psi}^*}(\mathbf{x}_t, y, t) = \mathbb{E}_{p(\mathbf{x}_0|\mathbf{x}_t,y)} \left[ \exp(\frac{1}{\beta} r(\mathbf{x}_0, y)) \right].$$

By Lemma 3.2, we can instead estimate the value $\mathbb{E}_{p(\mathbf{x}_0|\mathbf{x}_t,y)}\left[\exp(\frac{1}{\beta} r(\mathbf{x}_0, y))\right]$ using the guidance network $h_{\boldsymbol{\psi}^*}$ obtained by minimizing the objective function $\mathcal{L}_{\text{guidance}}(\boldsymbol{\psi})$, which can be approximated by easy sampling from the joint distribution $p(\mathbf{x}_0, \mathbf{x}_t, y)$. Then, the estimated score function for the aligned diffusion model can be calculated as follows:

$$\mathbf{s}_{\boldsymbol{\phi}^*}(\mathbf{x}_t, y, t) = \underbrace{\nabla_{\mathbf{x}_t} \log p(\mathbf{x}_t|y)}_{\substack{\text{pre-trained model} \\ \text{on source}}} + \underbrace{\nabla_{\mathbf{x}_t} \log h_{\boldsymbol{\psi}^*}(\mathbf{x}_t, y, t)}_{\text{guidance network}}. \tag{14}$$

To alleviate the adversarial nature of the guidance, we can adopt the consistency regularization $\mathcal{L}_{\text{consistence}}$ to learn the guidance network $h_{\boldsymbol{\psi}^*}$ better, i.e., the gradient of $\mathcal{L}_{\text{consistence}}(\mathbf{x}_t, y, t)$ with respect to $\mathbf{x}_t$ should match the score in preferred data. The key point of this regularization is that we

---

[1] Although there is no $\alpha$ in equation 19, many guidance methods (Lu et al., 2023; Song et al., 2023) add this hyperparameter in practice to balance the strength of guidance term with the score.

[2] We use landscape to describe the change of reward given the change of images.

Table 1: Comparison of finetuning-free alignment algorithms. Our method uniquely provides theoretical guarantees for the correct form for guidance with a step size guarantee.

| Method | Classifier Guidance | Direct backpropagate (M1) | Tweedie's formula (M2) | Ours |
|---|---|---|---|---|
| Formulation | $\frac{1}{\beta}\nabla_{\mathbf{x}_t} r(\mathbf{x}_t, y)$ | $\nabla_{\mathbf{x}_t} \log \frac{1}{n}\sum_{i=1}^{n} \exp\left(\frac{1}{\beta}r(\mathbf{x}_0^i, y)\right)$ | $\frac{1}{\beta}\nabla_{\mathbf{x}_t} r\left(\mathbb{E}_{p(\mathbf{x}_0 \mid \mathbf{x}_t, y)}[\mathbf{x}_0], y\right)$ | $\nabla_{\mathbf{x}_t} \log h_{\psi^*}(\mathbf{x}_t, y, t)$ |
| Unbiased | ✗ | ✓ | ✗ | ✓ |
| Step size guarantee | ✗ | ✗ | ✗ | ✓ |

cannot easily change the landscape of a given predetermined reward function, but we can regularize the landscape of the learned guidance network to ensure the generation of high-quality images.

$$\psi^* = \arg\min_{\psi} \mathcal{L}_{\text{consistence}}$$

$$:= \mathbb{E}_{q(\mathbf{x}_0, y)}\mathbb{E}_{q(\mathbf{x}_t \mid \mathbf{x}_0)}\left[\left\|\nabla_{\mathbf{x}_t}\log p(\mathbf{x}_t \mid \mathbf{x}_0, y) + \nabla_{\mathbf{x}_t}\log h_{\psi}(\mathbf{x}_t, y, t) - \nabla_{\mathbf{x}_t}\log q(\mathbf{x}_t \mid \mathbf{x}_0, y)\right\|_2^2\right].$$

$$(15)$$

Combining the consistency regularization terms together with the original guidance loss equation 24, the final learning objective for the guidance network can be described as follows:

$$\psi^* = \arg\min_{\psi}\{\mathcal{L}_{\text{guidance}} + \eta\,\mathcal{L}_{\text{consistence}}\}, \qquad (16)$$

where $\eta \geq 0$ are hyperparameters that control the strength of additional regularization, which also enhances the flexibility of our solution scheme.

### 3.4 FURTHER IMPROVEMENT TO ONE-STEP GENERATION

The training objective in equation 24 and equation 15 is agnostic to the reference model, meaning we can use any pre-trained diffusion model with any reward function, whether differentiable or not. Given that one-step generative models are fast and computationally efficient for practical use, can we design an explicit training objective for a one-step text-to-image model?

Surprisingly, the solution is remarkably simple—instead of sampling $t$ uniformly from $[0, T]$, we can simply set $t = T$. This small modification offers several advantages. First, while one-step diffusion models may not perform as well as few-step (2–4) models (Salimans & Ho, 2022), we empirically find that with additional guidance, their performance improves significantly, as presented in Section 4.3. Additionally, the guidance network $h_{\psi}$ becomes time-independent, meaning it always guides the diffusion model from $x_T$ to $x_0$. Empirically, we find that $h_{\psi}$ is easy to train—with ten training epochs on the Pick-a-Pic V1 dataset, our guidance network produces high-quality images, which can be found in Section 4.2. We summarize the final learning pipeline in Algorithm 1 in the Appendix.

### 3.5 FURTHER EXTENSION TO FLOW MATCHING

Given that state-of-the-art models are grounded in Diffusion Transformers (Peebles & Xie, 2022) and flow matching (Lipman et al., 2022), we present the exact form of flow-matching guidance in the theorem below.

**Theorem 3.3.** *Let $\phi^*$ be the optimal solution for the conditional flow matching model trained on target domain $q(\mathbf{x}_1, y)$ (where $\mathbf{x}_1$ are sampled from data distribution, $\mathbf{v}_q(\mathbf{x}_t, y, t)$ denotes the oracle velocity field on target distribution), i.e.,*

$$\phi_q^* = \arg\min_{\phi} \mathbb{E}_t\left\{\mathbb{E}_{q_t(\mathbf{x}_t, y)}\left[\left\|\mathbf{v}_{\phi}(\mathbf{x}_t, y, t) - \mathbf{v}_q(\mathbf{x}_t, y, t)\right\|_2^2\right]\right\},$$

*then*

$$\mathbf{v}_{\phi_q^*}(\mathbf{x}_t, y, t) = \mathbf{v}_{\phi_p}(\mathbf{x}_t, y, t) +$$

$$\mathbb{E}_{\mathbf{x}_1 \sim p_{1\mid t}(\mathbf{x}_1 \mid \mathbf{x}_t, y)}\left[\left(\frac{\exp\left(\frac{1}{\beta}r(\mathbf{x}_1, y)\right)}{\mathbb{E}_{\mathbf{x}_1' \sim p_{1\mid t}(\mathbf{x}_1 \mid \mathbf{x}_t, y)}\left[\exp\left(\frac{1}{\beta}r(\mathbf{x}_1', y)\right)\right]} - 1\right)\mathbf{v}_t(\mathbf{x}_t \mid \mathbf{x}_1, y)\right], \qquad (17)$$

According to the formulation of equation 17, we propose a training-free guidance that directly calculates the guidance term, which can be found in A.2. Compared with no base-model fine-tuning proposed in equation 14, this formulation offers greater computational efficiency.

## 4 EXPERIMENTAL RESULTS

In this section, we present our comprehensive experimental evaluation, demonstrating the effectiveness of our finetuning-free method for sampling directly from reward-guided distributions. We first outline our experimental setup and evaluation criteria in Section 4.1, followed by benchmark results against state-of-the-art methods in Section 4.2. Finally, we provide an in-depth ablation study that validates our key theoretical claims and demonstrates the superior performance of our guidance network in Section 4.3.

### 4.1 EXPERIMENTAL SETUP

We follow the official configurations recommended for SPO (Liang et al., 2024), Diffusion-DPO (Wallace et al., 2023), and MAPO (She et al., 2024). Diffusion-DPO and MAPO are fine-tuned on the Pick-a-Pic V2 dataset, which contains over 800k image preference pairs. In contrast, SPO is fine-tuned online using 4k text prompts (without images) randomly selected from Pick-a-Pic V1. Our method trains the guidance network offline using 583k image preference pairs from Pick-a-Pic V1. Overall, our method and the competing models in the text-to-image alignment benchmark are trained on comparable datasets, allowing for a fair comparison.

We adopt Stable Diffusion XL (SDXL)-Turbo as the reference model for one-step text-to-image generation. SDXL-Turbo is specifically designed for fast inference by incorporating a distillation process that reduces the number of required denoising steps while maintaining high image quality. Unlike SDXL, which typically requires 20–50 steps for high-fidelity generation, SDXL-Turbo leverages progressive distillation (Sauer et al., 2023) to achieve comparable performance in as few as one to four steps. Since the distillation dataset is a subset of the original SDXL training data, it may not introduce additional information to improve performance. We also include ablation studies in Section 4.3 to verify the effectiveness of our method.

**Implementation Detail** Since the guidance network takes noisy images $\mathbf{x}_T$ and prompts $y$ as input and outputs a scalar value, we adopt the same variational autoencoder (VAE), tokenizer, and text encoder from the reference diffusion model for encoding image and text. Consequently, the trainable parameters of our guidance network are quite small. In practice, we adopt two convolutional layers for processing VAE-encoded feature maps and a five-layer multi-layer perceptron (MLP) to project the image and text embedding to a scalar. The total parameter size of the guidance network is only 72 MB, making it lightweight and easy to train. We train the guidance network on the Pick-a-Pic training dataset for 10 epochs with batch size 32, Adam optimizer, learning rate 1e-3, and hyperparameters $\eta = 1$.

**Evaluation Criterion** Following established evaluation protocols (Wallace et al., 2023; Liang et al., 2024), we report quantitative results using 500 validation prompts from the validation unique split of Pick-a-Pic. We adopt four evaluation criteria to evaluate different aspects of image quality. PickScore (Kirstain et al., 2023) measures overall human preference by aggregating judgments on aesthetic appeal, coherence, and realism. HPSV2 (Wu et al., 2023) assesses prompt adherence, ensuring the generated image accurately reflects the given textual description. ImageReward (Xu et al., 2023) quantifies human preference based on fine-grained attributes such as composition, detail preservation, and semantic relevance. Lastly, the aesthetic evaluation model from LAION (Schuhmann, 2022) focuses on visual appeal, capturing factors such as color harmony, style, and artistic quality.

### 4.2 EXPERIMENTAL RESULTS

As shown in Table 2, our method surpasses baseline approaches across four evaluation criteria, demonstrating its effectiveness in enhancing text-to-image alignment. The improvements are observed in both perceptual quality and semantic coherence, indicating that our guidance network

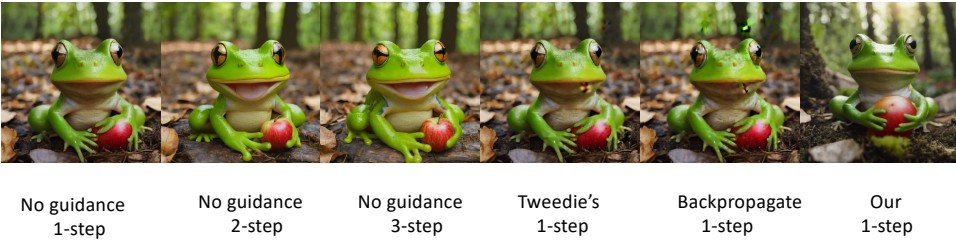

| No guidance
1-step | No guidance
2-step | No guidance
3-step | Tweedie's
1-step | Backpropagate
1-step | Our
1-step |

Figure 3: Effectiveness of the Proposed Method: The results demonstrate that 2-step and 3-step generation significantly improve the quality of the generated images compared to one-step generation. While two vanilla guidance methods (Tweedie's formula or directly backpropagation summarized in Section 3.2) fail to produce meaningful changes in the scene despite appropriate guidance strength, our method successfully achieves this enhancement. The prompt is "A photo of a frog holding an apple while smiling in the forest".

Table 2: Benchmark comparison of different methods on text-to-image alignment. Our method achieves the highest scores in PickScore, HPSV2, and Aesthetic metrics, demonstrating superior alignment and image quality.

| Type | Method | PickScore | HPSV2 | ImageReward | Aesthetic | Training GPU Hour |
|---|---|---|---|---|---|---|
| Baseline | SDXL | 21.95 | 26.95 | 0.5380 | 5.950 | - |
| Training-free | Direct backpropagate | 21.84 | 27.53 | 0.5870 | 5.922 | - |
| | Tweedie's formula | 22.34 | 28.76 | 0.9501 | 6.002 | - |
| Finetuning-based | Diff.-DPO | 22.64 | 29.31 | 0.9436 | 6.015 | 4800 |
| | SPO | 23.06 | 31.80 | **1.0803** | 6.364 | 234 |
| Finetuning-free | Ours | **23.08** | **32.12** | 1.0625 | **6.452** | **92** |
| Baseline | SD3.5 large turbo | 22.30 | 30.29 | 1.0159 | 6.5190 | - |
| Finetuning-free | Ours | **23.14** | **32.31** | 1.1025 | **6.5280** | - |

successfully refines image generation to better match textual descriptions. This performance gain highlights the advantages of our lightweight architecture and the optimization strategy used during training. Figure 1 provides a qualitative comparison with baseline methods, further illustrating the superior visual fidelity and text alignment achieved by our approach.

## 4.3 ABLATION STUDY

In this section, we first verify the advantages of our proposed method against other finetuning-free guidance methods in Table 1. We then analyze the impact of few-step (2–4) generation compared to one-step generation, highlighting how our guidance term significantly enhances performance.

As illustrated in Figure 3, vanilla guidance methods struggle to induce meaningful improvements in generated images, even with carefully tuned guidance strength. Increasing the guidance parameter $\alpha$ often leads to undesirable artifacts rather than quality improvements. In contrast, our method effectively enhances image generation by leveraging a regularized guidance network, demonstrating its ability to refine scene details and improve alignment with input prompts.

Figure 4: Ablation study comparing the performance of our method with no guidance and two vanilla guidance methods under one-step and multi-step generation. Our method outperforms all baselines, which demonstrates the effectiveness of our guidance network in refining image quality and prompt alignment.

| Method | PickScore |
|---|---|
| Ours (1 step) | **23.08** |
| No guidance (1 step) | 22.14 |
| Tweedie's (1 step) | 22.34 |
| Backpropagate (1 step) | 21.84 |
| No guidance (2 steps) | 22.64 |
| No guidance (3 steps) | 22.56 |

To further explore this, we examine the performance of our method against two vanilla guidance techniques, Tweedie's and Backpropagate, as well as the no guidance baseline, all under a one-step sampling condition. As shown in Table 4, our method achieves the highest PickScore. This demonstrates that our regularized guidance network provides a substantial improvement over no guidance scenario and traditional methods. Consistent with prior studies, increasing the number of steps from one to two or three results in improved image quality, as shown in Figure 3 and Table 4. However, our method enables one-step generation to achieve performance even better than 2- or 3-step generation, highlighting the power of our guidance network. In Appendix B, we include the sensitive analysis of the regularization strength and we provide the experiments with non-differentiable reward on the GenEval benchmark (Ghosh et al., 2023) in Appendix B.4.

## 5 CONCLUSION

In this paper, we introduced a novel finetuning-free framework for aligning text-to-image diffusion models with human preferences. By formulating alignment as sampling from a reward-weighted distribution, our approach eliminates the need for computationally expensive fine-tuning and instead leverages a plug-and-play guidance mechanism. Specifically, we decomposed the score function into a pre-trained score and an additional guidance term, enabling efficient alignment without modifying the underlying diffusion model. Moreover, we identified a key challenge: the adversarial nature of the guidance term can lead to undesirable artifacts. To mitigate this, we proposed a regularization strategy that stabilizes guidance. Our experimental results on the text-to-image benchmark demonstrated that our method effectively aligns model outputs with human preferences.

## ETHICS STATEMENT

This work complies with the ICLR Code of Ethics. It does not involve human subjects, sensitive personal information, or experiments with potential harm to individuals or communities. All experiments rely exclusively on publicly available datasets and benchmarks.

## REPRODUCIBILITY STATEMENT

We attach the code in the supplementary materials. The assumption and theoretical result can be found in Theorem A.1. The proof can be found in Appendix A.1.

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

# A    THEORETICAL DETAILS FOR SECTION 3

## A.1    PROOF OF THEOREM A.1

We first provide the formal theorem as follows:

**Theorem A.1.** *Let the conditional distribution of reference diffusion model $\pi_{ref}(\mathbf{x}|y)$ be denoted as distribution $p$ and the reward-weighted distribution $\pi_r(\mathbf{x}|y)$ defined in equation 6 as distribution $q$. Assume $\mathbf{x}_t$ and $y$ are conditionally independent given $\mathbf{x}_0$ in the forward process, i.e., $p(\mathbf{x}_t|\mathbf{x}_0, y) = p(\mathbf{x}_t|\mathbf{x}_0)$, $\forall t \in [0, T]$. Additionally, assume the forward process on the reward-weighted distribution is identical to that on the reference distribution $q(\mathbf{x}_t|\mathbf{x}_0) = p(\mathbf{x}_t|\mathbf{x}_0)$[3], and $\phi^*$ is the optimal solution for the conditional diffusion model trained on target domain $q(\mathbf{x}_0, y)$, i.e.,*

$$\phi^* = \arg \min_{\phi} \mathbb{E}_t \left\{ \lambda(t) \mathbb{E}_{q_t(\mathbf{x}_t, y)} \left[ \left\| \mathbf{s}_\phi(\mathbf{x}_t, y, t) - \nabla_{\mathbf{x}_t} \log q_t(\mathbf{x}_t|y) \right\|_2^2 \right] \right\}, \tag{18}$$

*then*

$$\mathbf{s}_{\phi^*}(\mathbf{x}_t, y, t) = \underbrace{\nabla_{\mathbf{x}_t} \log p_t(\mathbf{x}_t|y)}_{\substack{\text{pre-trained conditional model} \\ \text{on source}}} + \underbrace{\nabla_{\mathbf{x}_t} \log \mathbb{E}_{p(\mathbf{x}_0|\mathbf{x}_t, y)} \left[ \exp(\frac{1}{\beta} r(\mathbf{x}_0, y)) \right]}_{\text{conditional guidance}}. \tag{19}$$

*Proof.* The proof is based on the theoretical framework of Ouyang et al. (2024). For the ease of readers, we incorporate the relevant conclusion from their work as lemmas below. To prove Eq 19, we first build the connection between the Conditional Score Matching on the target domain and Importance Weighted Conditional Denoising Score Matching on the source domain in the following Lemma:

**Lemma A.2.** *Conditional Score Matching on the target domain is equivalent to Importance Weighted Denoising Score Matching on the source domain, i.e.,*

$$\phi^* = \arg \min_{\phi} \mathbb{E}_t \left\{ \lambda(t) \mathbb{E}_{q_t(\mathbf{x}_t, y)} \left[ \| \mathbf{s}_\phi(\mathbf{x}_t, y, t) - \nabla_{\mathbf{x}_t} \log q_t(\mathbf{x}_t|y) \|_2^2 \right] \right\}$$

$$= \arg \min_{\phi} \mathbb{E}_t \left\{ \lambda(t) \mathbb{E}_{p(\mathbf{x}_0, y)} \mathbb{E}_{p(\mathbf{x}_t|\mathbf{x}_0)} \left[ \| \mathbf{s}_\phi(\mathbf{x}_t, y, t) - \nabla_{\mathbf{x}_t} \log p(\mathbf{x}_t|\mathbf{x}_0) \|_2^2 \frac{q(\mathbf{x}_0, y)}{p(\mathbf{x}_0, y)} \right] \right\}.$$

*Proof of Lemma A.2.* We first connect the Conditional Score Matching objective in the target domain to the Conditional Denoising Score Matching objective in target distribution, which is proven by Batzolis et al. (2021), i.e.,

$$\phi^* = \arg \min_{\phi} \mathbb{E}_t \left\{ \lambda(t) \mathbb{E}_{q_t(\mathbf{x}_t, y)} \left[ \| \mathbf{s}_\phi(\mathbf{x}_t, y, t) - \nabla_{\mathbf{x}_t} \log q_t(\mathbf{x}_t|y) \|_2^2 \right] \right\}$$

$$= \arg \min_{\phi} \mathbb{E}_t \left\{ \lambda(t) \mathbb{E}_{q(\mathbf{x}_0, y)} \mathbb{E}_{q(\mathbf{x}_t|\mathbf{x}_0)} \left[ \| \mathbf{s}_\phi(\mathbf{x}_t, y, t) - \nabla_{\mathbf{x}_t} \log q(\mathbf{x}_t|\mathbf{x}_0) \|_2^2 \right] \right\}.$$

Then we split the mean squared error of the Conditional Denoising Score Matching objective on the target distribution into three terms as follows:

$$\mathbb{E}_{q(\mathbf{x}_0, y)} \mathbb{E}_{q(\mathbf{x}_t|\mathbf{x}_0)} \left[ \| \mathbf{s}_\phi(\mathbf{x}_t, y, t) - \nabla_{\mathbf{x}_t} \log q(\mathbf{x}_t|\mathbf{x}_0) \|_2^2 \right]$$

$$= \mathbb{E}_{q(\mathbf{x}_0, \mathbf{x}_t, y)} \left[ \| \mathbf{s}_\phi(\mathbf{x}_t, y, t) \|_2^2 \right] - 2 \mathbb{E}_{q(\mathbf{x}_0, \mathbf{x}_t, y)} \left[ \langle \mathbf{s}_\phi(\mathbf{x}_t, y, t), \nabla_{\mathbf{x}_t} \log q(\mathbf{x}_t|\mathbf{x}_0) \rangle \right] + C_1, \tag{20}$$

where $C_1 = \mathbb{E}_{q(\mathbf{x}_0, \mathbf{x}_t, y)} \left[ \| \nabla_{\mathbf{x}_t} \log q(\mathbf{x}_t|\mathbf{x}_0) \|_2^2 \right]$ is a constant independent with $\phi$, and $q(\mathbf{x}_t|\mathbf{x}_0, y) = q(\mathbf{x}_t|\mathbf{x}_0)$ because of conditional independent of $\mathbf{x}_t$ and $y$ given $\mathbf{x}_0$ by assumption. We can similarly

---

[3]These two assumptions are mild since $\mathbf{x}_0$ contains all information about $y$ and $p(\mathbf{x}_t|\mathbf{x}_0)$ and $q(\mathbf{x}_t|\mathbf{x}_0)$ are forward noising process, which is easy to control.

split the mean squared error of Denoising Score Matching on the source domain into three terms as follows:

$$\mathbb{E}_{p(\mathbf{x}_0,y)}\mathbb{E}_{p(\mathbf{x}_t|\mathbf{x}_0)}\left[\|\mathbf{s}_\phi(\mathbf{x}_t,y,t)-\nabla_{\mathbf{x}_t}\log p(\mathbf{x}_t|\mathbf{x}_0)\|_2^2\frac{q(\mathbf{x}_0,y)}{p(\mathbf{x}_0,y)}\right]$$

$$=\mathbb{E}_{p(\mathbf{x}_0,\mathbf{x}_t,y)}\left[\|\mathbf{s}_\phi(\mathbf{x}_t,y,t)\|_2^2\frac{q(\mathbf{x}_0,y)}{p(\mathbf{x}_0,y)}\right]-2\mathbb{E}_{p(\mathbf{x}_0,\mathbf{x}_t,y)}\left[\langle\mathbf{s}_\phi(\mathbf{x}_t,y,t),\nabla_{\mathbf{x}_t}\log p(\mathbf{x}_t|\mathbf{x}_0)\rangle\frac{q(\mathbf{x}_0,y)}{p(\mathbf{x}_0,y)}\right]$$

$$+C_2,$$

(21)

where $C_2$ is a constant independent with $\phi$.

It is obvious to show that the first term in equation 20 is equal to the first term in equation 21, i.e.,

$$\mathbb{E}_{p(\mathbf{x}_0,\mathbf{x}_t,y)}\left[\|\mathbf{s}_\phi(\mathbf{x}_t,y,t)\|_2^2\frac{q(\mathbf{x}_0,y)}{p(\mathbf{x}_0,y)}\right]$$

$$=\int_{\mathbf{x}_0}\int_{\mathbf{x}_t}\int_y p(\mathbf{x}_0,y)p(\mathbf{x}_t|\mathbf{x}_0)\|\mathbf{s}_\phi(\mathbf{x}_t,y,t)\|_2^2\frac{q(\mathbf{x}_0,y)}{p(\mathbf{x}_0,y)}d\mathbf{x}_0 d\mathbf{x}_t dy$$

$$=\int_{\mathbf{x}_0}\int_{\mathbf{x}_t}\int_y p(\mathbf{x}_0,y)q(\mathbf{x}_t|\mathbf{x}_0)\|\mathbf{s}_\phi(\mathbf{x}_t,y,t)\|_2^2\frac{q(\mathbf{x}_0,y)}{p(\mathbf{x}_0,y)}d\mathbf{x}_0 d\mathbf{x}_t dy$$

$$=\int_{\mathbf{x}_0}\int_{\mathbf{x}_t}\int_y q(\mathbf{x}_0,\mathbf{x}_t,y)\|\mathbf{s}_\phi(\mathbf{x}_t,y,t)\|_2^2 d\mathbf{x}_0 d\mathbf{x}_t dy$$

$$=\mathbb{E}_{q(\mathbf{x}_0,\mathbf{x}_t,y)}\left[\|\mathbf{s}_\phi(\mathbf{x}_t,y,t)\|_2^2\right].$$

And the second term is also equivalent:

$$\mathbb{E}_{p(\mathbf{x}_0,\mathbf{x}_t,y)}\left[\langle\mathbf{s}_\phi(\mathbf{x}_t,y,t),\nabla_{\mathbf{x}_t}\log p(\mathbf{x}_t|\mathbf{x}_0)\rangle\frac{q(\mathbf{x}_0,y)}{p(\mathbf{x}_0,y)}\right]$$

$$=\int_{\mathbf{x}_0}\int_{\mathbf{x}_t}\int_y p(\mathbf{x}_0,\mathbf{x}_t,y)\langle\mathbf{s}_\phi(\mathbf{x}_t,y,t),\frac{\nabla_{\mathbf{x}_t}p(\mathbf{x}_t|\mathbf{x}_0)}{p(\mathbf{x}_t|\mathbf{x}_0)}\rangle\frac{q(\mathbf{x}_0,y)}{p(\mathbf{x}_0,y)}d\mathbf{x}_0 d\mathbf{x}_t dy$$

$$=\int_{\mathbf{x}_0}\int_{\mathbf{x}_t}\int_y p(\mathbf{x}_0,\mathbf{x}_t,y)\langle\mathbf{s}_\phi(\mathbf{x}_t,y,t),\frac{\nabla_{\mathbf{x}_t}q(\mathbf{x}_t|\mathbf{x}_0)}{p(\mathbf{x}_t|\mathbf{x}_0)}\rangle\frac{q(\mathbf{x}_0,y)}{p(\mathbf{x}_0,y)}d\mathbf{x}_0 d\mathbf{x}_t dy$$

$$=\int_{\mathbf{x}_0}\int_{\mathbf{x}_t}\int_y \langle\mathbf{s}_\phi(\mathbf{x}_t,y,t),\nabla_{\mathbf{x}_t}q(\mathbf{x}_t|\mathbf{x}_0)\rangle q(\mathbf{x}_0,y)d\mathbf{x}_0 d\mathbf{x}_t dy$$

$$=\int_{\mathbf{x}_0}\int_{\mathbf{x}_t}\int_y \langle\mathbf{s}_\phi(\mathbf{x}_t,y,t),\nabla_{\mathbf{x}_t}\log q(\mathbf{x}_t|\mathbf{x}_0)\rangle q(\mathbf{x}_t|\mathbf{x}_0)q(\mathbf{x}_0,y)d\mathbf{x}_0 d\mathbf{x}_t dy$$

$$=\mathbb{E}_{q(\mathbf{x}_0,\mathbf{x}_t,y)}\left[\langle\mathbf{s}_\phi(\mathbf{x}_t,y,t),\nabla_{\mathbf{x}_t}\log q(\mathbf{x}_t|\mathbf{x}_0)\rangle\right].$$

$\square$

**Lemma A.3.** *Assume* $\mathbf{x}_t$ *and* $y$ *are conditional independent given* $\mathbf{x}_0$ *in the forward process, i.e.,* $p(\mathbf{x}_t|\mathbf{x}_0,y)=p(\mathbf{x}_t|\mathbf{x}_0)$, $\forall t\in[0,T]$, *and let the forward process on the target domain be identical to that on the source domain* $q(\mathbf{x}_t|\mathbf{x}_0)=p(\mathbf{x}_t|\mathbf{x}_0)$, *and* $\phi^*$ *is the optimal solution for the conditional diffusion model trained on target domain* $q(\mathbf{x}_0,y)$, *i.e.,*

$$\phi^*=\arg\min_\phi\mathbb{E}_t\left\{\lambda(t)\mathbb{E}_{q_t(\mathbf{x}_t,y)}\left[\|\mathbf{s}_\phi(\mathbf{x}_t,y,t)-\nabla_{\mathbf{x}_t}\log q_t(\mathbf{x}_t|y)\|_2^2\right]\right\},\qquad(22)$$

*then*

$$\mathbf{s}_{\phi^*}(\mathbf{x}_t,y,t)=\nabla_{\mathbf{x}_t}\log p_t(\mathbf{x}_t|y)+\nabla_{\mathbf{x}_t}\log\mathbb{E}_{p(\mathbf{x}_0|\mathbf{x}_t,y)}\left[\frac{q(\mathbf{x}_0,y)}{p(\mathbf{x}_0,y)}\right].\qquad(23)$$

*Proof of Lemma A.3.* According to Lemma A.2, the optimal solution satisfies

$$\phi^*=\arg\min_\phi\mathbb{E}_t\left\{\lambda(t)\mathbb{E}_{p(\mathbf{x}_0,y)}\mathbb{E}_{p(\mathbf{x}_t|\mathbf{x}_0)}\left[\|\mathbf{s}_\phi(\mathbf{x}_t,y,t)-\nabla_{\mathbf{x}_t}\log p(\mathbf{x}_t|\mathbf{x}_0)\|_2^2\frac{q(\mathbf{x}_0,y)}{p(\mathbf{x}_0,y)}\right]\right\}$$

where $Z(y) = \int p(\mathbf{x}_0, y) \exp\left(\frac{1}{\beta} r(\mathbf{x}_0, y)\right) d\mathbf{x}$. Then, we use Importance Weighted Conditional Denoising Score Matching on the source domain to get the analytic form of $\mathbf{s}_{\phi^*}$ as follows:

$$\mathbf{s}_{\phi^*}(\mathbf{x}_t, y, t) = \frac{\mathbb{E}_{p(\mathbf{x}_0|\mathbf{x}_t, y)}\left[\nabla_{\mathbf{x}_t} \log p(\mathbf{x}_t|\mathbf{x}_0) \frac{q(\mathbf{x}_0, y)}{p(\mathbf{x}_0, y)}\right]}{\mathbb{E}_{p(\mathbf{x}_0|\mathbf{x}_t, y)}\left[\frac{q(\mathbf{x}_0, y)}{p(\mathbf{x}_0, y)}\right]}.$$

Moreover, the RHS of equation 23 can be rewritten as:

$$\text{RHS} = \nabla_{\mathbf{x}_t} \log p_t(\mathbf{x}_t|y) + \nabla_{\mathbf{x}_t} \log \mathbb{E}_{p(\mathbf{x}_0|\mathbf{x}_t, y)}\left[\frac{q(\mathbf{x}_0, y)}{p(\mathbf{x}_0, y)}\right]$$

$$= \nabla_{\mathbf{x}_t} \log p_t(\mathbf{x}_t|y) + \frac{\nabla_{\mathbf{x}_t} \mathbb{E}_{p(\mathbf{x}_0|\mathbf{x}_t, y)}\left[\frac{q(\mathbf{x}_0, y)}{p(\mathbf{x}_0, y)}\right]}{\mathbb{E}_{p(\mathbf{x}_0|\mathbf{x}_t, y)}\left[\frac{q(\mathbf{x}_0, y)}{p(\mathbf{x}_0, y)}\right]}$$

$$= \nabla_{\mathbf{x}_t} \log p_t(\mathbf{x}_t|y) + \frac{\mathbb{E}_{p(\mathbf{x}_0|\mathbf{x}_t, y)}\left[\frac{q(\mathbf{x}_0, y)}{p(\mathbf{x}_0, y)} \nabla_{\mathbf{x}_t} \log p(\mathbf{x}_0|\mathbf{x}_t, y)\right]}{\mathbb{E}_{p(\mathbf{x}_0|\mathbf{x}_t, y)}\left[\frac{q(\mathbf{x}_0, y)}{p(\mathbf{x}_0, y)}\right]}.$$

Since

$$\nabla_{\mathbf{x}_t} \log p(\mathbf{x}_0|\mathbf{x}_t, y) = \nabla_{\mathbf{x}_t} \log p(\mathbf{x}_t|\mathbf{x}_0, y) + \nabla_{\mathbf{x}_t} \log p(\mathbf{x}_0|y) - \nabla_{\mathbf{x}_t} \log p_t(\mathbf{x}_t|y)$$
$$= \nabla_{\mathbf{x}_t} \log p(\mathbf{x}_t|\mathbf{x}_0, y) - \nabla_{\mathbf{x}_t} \log p_t(\mathbf{x}_t|y),$$
$$= \nabla_{\mathbf{x}_t} \log p(\mathbf{x}_t|\mathbf{x}_0) - \nabla_{\mathbf{x}_t} \log p_t(\mathbf{x}_t|y),$$

we can further simplify the RHS of equation 23 as follows:

$$\text{RHS} = \nabla_{\mathbf{x}_t} \log p_t(\mathbf{x}_t|y) + \frac{\mathbb{E}_{p(\mathbf{x}_0|\mathbf{x}_t, y)}\left[\frac{q(\mathbf{x}_0, y)}{p(\mathbf{x}_0, y)} \nabla_{\mathbf{x}_t} \log p(\mathbf{x}_t|\mathbf{x}_0)\right]}{\mathbb{E}_{p(\mathbf{x}_0|\mathbf{x}_t, y)}\left[\frac{q(\mathbf{x}_0, y)}{p(\mathbf{x}_0, y)}\right]} - \nabla_{\mathbf{x}_t} \log p_t(\mathbf{x}_t|y)$$

$$= \frac{\mathbb{E}_{p(\mathbf{x}_0|\mathbf{x}_t, y)}\left[\nabla_{\mathbf{x}_t} \log p(\mathbf{x}_t|\mathbf{x}_0) \frac{q(\mathbf{x}_0, y)}{p(\mathbf{x}_0, y)}\right]}{\mathbb{E}_{p(\mathbf{x}_0|\mathbf{x}_t, y)}\left[\frac{q(\mathbf{x}_0, y)}{p(\mathbf{x}_0, y)}\right]}$$

$$= \mathbf{s}_{\phi^*}(\mathbf{x}_t, t).$$

Thereby, we finish the proof. $\square$

According to the lemma A.3, we replace the density ratio $\frac{q(\mathbf{x}_0, y)}{p(\mathbf{x}_0, y)}$ by $\frac{\exp\left(\frac{1}{\beta} r(\mathbf{x}_0, y)\right)}{Z(y)}$, we get

$$\mathbf{s}_{\phi^*}(\mathbf{x}_t, y, t) = \nabla_{\mathbf{x}_t} \log p_t(\mathbf{x}_t|y) + \nabla_{\mathbf{x}_t} \log \mathbb{E}_{p(\mathbf{x}_0|\mathbf{x}_t, y)}\left[\frac{q(\mathbf{x}_0, y)}{p(\mathbf{x}_0, y)}\right]$$

$$= \nabla_{\mathbf{x}_t} \log p_t(\mathbf{x}_t|y) + \nabla_{\mathbf{x}_t} \log \mathbb{E}_{p(\mathbf{x}_0|\mathbf{x}_t, y)}\left[\frac{\exp\left(\frac{1}{\beta} r(\mathbf{x}_0, y)\right)}{Z(y)}\right]$$

$$= \nabla_{\mathbf{x}_t} \log p_t(\mathbf{x}_t|y) + \nabla_{\mathbf{x}_t} \log \mathbb{E}_{p(\mathbf{x}_0|\mathbf{x}_t, y)}\left[\exp\left(\frac{1}{\beta} r(\mathbf{x}_0, y)\right)\right]$$

Thereby, we finish the proof. $\square$

## A.2 PROOF OF THEOREM 3.3

We provide the detailed discussion about training-free guidance of flow matching in this subsection.

*Proof of Theorem 3.3.* Denote $\boldsymbol{v}_t(\mathbf{x}_t, y)$ and $\boldsymbol{v}_t(\mathbf{x}_t \mid \mathbf{x}_1, y)$ as the marginal and conditional velocities, respectively. Then we have

$$\boldsymbol{v}_t^q(\mathbf{x}_t, y) = \mathbb{E}_{\mathbf{x}_1 \sim q_{1|t}(\mathbf{x}_1|\mathbf{x}_t,y)} \left[ \boldsymbol{v}_t(\mathbf{x}_t \mid \mathbf{x}_1, y) \right]$$

$$= \mathbb{E}_{\mathbf{x}_1 \sim p_{1|t}(\mathbf{x}_1|\mathbf{x}_t,y)} \left[ \boldsymbol{v}_t(\mathbf{x}_t \mid \mathbf{x}_1, y) \frac{q_{1|t}(\mathbf{x}_1 \mid \mathbf{x}_t, y)}{p_{1|t}(\mathbf{x}_1 \mid \mathbf{x}_t, y)} \right]$$

$$= \mathbb{E}_{\mathbf{x}_1 \sim p_{1|t}(\mathbf{x}_1|\mathbf{x}_t,y)} \left[ \boldsymbol{v}_t(\mathbf{x}_t \mid \mathbf{x}_1, y) \frac{\frac{q_{t|1}(\mathbf{x}_t|\mathbf{x}_1,y)\,q_1(\mathbf{x}_1)}{q_t(\mathbf{x}_t,y)}}{\frac{p_{t|1}(\mathbf{x}_t|\mathbf{x}_1,y)\,p_1(\mathbf{x}_1)}{p_t(\mathbf{x}_t,y)}} \right]$$

$$= \mathbb{E}_{\mathbf{x}_1 \sim p_{1|t}(\mathbf{x}_1|\mathbf{x}_t,y)} \left[ \boldsymbol{v}_t(\mathbf{x}_t \mid \mathbf{x}_1, y) \frac{q_{t|1}(\mathbf{x}_t \mid \mathbf{x}_1, y)\,q_1(\mathbf{x}_1)\,p_t(\mathbf{x}_t, y)}{p_{t|1}(\mathbf{x}_t \mid \mathbf{x}_1, y)\,p_1(\mathbf{x}_1)\,q_t(\mathbf{x}_t, y)} \right]$$

$$= \mathbb{E}_{\mathbf{x}_1 \sim p_{1|t}(\mathbf{x}_1|\mathbf{x}_t,y)} \left[ \boldsymbol{v}_t(\mathbf{x}_t \mid \mathbf{x}_1, y) \frac{q_1(\mathbf{x}_1)}{p_1(\mathbf{x}_1)} \cdot \frac{p_t(\mathbf{x}_t, y)}{q_t(\mathbf{x}_t, y)} \right] \qquad \text{(because } q_{t|1}(\mathbf{x}_t \mid \mathbf{x}_1, y) = p_{t|1}(\mathbf{x}_t \mid \mathbf{x}_1, y))$$

$$= \mathbb{E}_{\mathbf{x}_1 \sim p_{1|t}(\mathbf{x}_1|\mathbf{x}_t,y)} \left[ \boldsymbol{v}_t(\mathbf{x}_t \mid \mathbf{x}_1, y) \frac{\frac{q_1(\mathbf{x}_1)}{p_1(\mathbf{x}_1)}}{\frac{q_t(\mathbf{x}_t,y)}{p_t(\mathbf{x}_t,y)}} \right]$$

$$= \mathbb{E}_{\mathbf{x}_1 \sim p_{1|t}(\mathbf{x}_1|\mathbf{x}_t,y)} \left[ \boldsymbol{v}_t(\mathbf{x}_t \mid \mathbf{x}_1, y) \frac{\frac{q_1(\mathbf{x}_1)}{p_1(\mathbf{x}_1)}}{\sum_{\mathbf{x}_1'} p_{1|t}(\mathbf{x}_1' \mid \mathbf{x}_t, y) \frac{q_1(\mathbf{x}_1')}{p_1(\mathbf{x}_1')}} \right]$$

$$= \mathbb{E}_{\mathbf{x}_1 \sim p_{1|t}(\mathbf{x}_1|\mathbf{x}_t,y)} \left[ \boldsymbol{v}_t(\mathbf{x}_t \mid \mathbf{x}_1, y) \frac{\frac{q_1(\mathbf{x}_1)}{p_1(\mathbf{x}_1)}}{\mathbb{E}_{\mathbf{x}_1' \sim p_{1|t}(\mathbf{x}_1|\mathbf{x}_t,y)} \left[ \frac{q_1(\mathbf{x}_1')}{p_1(\mathbf{x}_1')} \right]} \right]$$

$$= \mathbb{E}_{\mathbf{x}_1 \sim p_{1|t}(\mathbf{x}_1|\mathbf{x}_t,y)} \left[ \boldsymbol{v}_t(\mathbf{x}_t \mid \mathbf{x}_1, y) \frac{\exp\left(\frac{1}{\beta} r(\mathbf{x}_1, y)\right)}{\mathbb{E}_{\mathbf{x}_1' \sim p_{1|t}(\mathbf{x}_1|\mathbf{x}_t,y)} \left[ \exp\left(\frac{1}{\beta} r(\mathbf{x}_1', y)\right) \right]} \right]$$

$$= \mathbb{E}_{\mathbf{x}_1 \sim p_{1|t}(\mathbf{x}_1|\mathbf{x}_t,y)} \left[ \boldsymbol{v}_t(\mathbf{x}_t \mid \mathbf{x}_1, y) \frac{\exp\left(\frac{1}{\beta} r(\mathbf{x}_1, y)\right)}{\mathbb{E}_{\mathbf{x}_1' \sim p_{1|t}(\mathbf{x}_1|\mathbf{x}_t,y)} \left[ \exp\left(\frac{1}{\beta} r(\mathbf{x}_1', y)\right) \right]} \right]$$

$$= \boldsymbol{v}_t^p(\mathbf{x}_t, y) + \mathbb{E}_{\mathbf{x}_1 \sim p_{1|t}(\mathbf{x}_1|\mathbf{x}_t,y)} \left[ \left( \frac{\exp\left(\frac{1}{\beta} r(\mathbf{x}_1, y)\right)}{\mathbb{E}_{\mathbf{x}_1' \sim p_{1|t}(\mathbf{x}_1|\mathbf{x}_t,y)} \left[ \exp\left(\frac{1}{\beta} r(\mathbf{x}_1', y)\right) \right]} - 1 \right) \boldsymbol{v}_t(\mathbf{x}_t \mid \mathbf{x}_1, y) \right].$$

The above derivation is the training-based guidance for flow matching, where we need to train the first guidance network $\boldsymbol{\psi}_1^*$ satisfies:

$$h_{\boldsymbol{\psi}_1^*}(\mathbf{x}_t, y, t) = \mathbb{E}_{\mathbf{x}_1 \sim p_{1|t}(\mathbf{x}_1|\mathbf{x}_t,y)} \left[ \exp\left(\frac{1}{\beta} r(\mathbf{x}_1, y)\right) \right]$$

by minimizing the objective

$$\mathcal{L}_{\text{guidance}}(\boldsymbol{\psi}_1) := \mathbb{E}_{p(\mathbf{x}_1, \mathbf{x}_t, y)} \left[ \left\| h_{\boldsymbol{\psi}_1}(\mathbf{x}_t, y, t) - \exp(\frac{1}{\beta} r(\mathbf{x}_1, y)) \right\|_2^2 \right].$$

And then we need the second guidance network $\boldsymbol{\psi}_2^*$ satisfies:

$$h_{\boldsymbol{\psi}_2^*}(\mathbf{x}_t, y, t) = \mathbb{E}_{\mathbf{x}_1 \sim p_{1|t}(\mathbf{x}_1|\mathbf{x}_t,y)} \left[ \left( \frac{\exp\left(\frac{1}{\beta} r(\mathbf{x}_1, y)\right)}{\mathbb{E}_{\mathbf{x}_1' \sim p_{1|t}(\mathbf{x}_1|\mathbf{x}_t,y)} \left[ \exp\left(\frac{1}{\beta} r(\mathbf{x}_1', y)\right) \right]} - 1 \right) \boldsymbol{v}_t(\mathbf{x}_t \mid \mathbf{x}_1, y) \right]$$

by minimizing the objective

$$\mathcal{L}_{\text{guidance}}(\boldsymbol{\psi}_2) := \mathbb{E}_{p(\mathbf{x}_1, \mathbf{x}_t, y)} \left[ \left\| h_{\boldsymbol{\psi}_2}(\mathbf{x}_t, y, t) - \left( \frac{\exp\left(\frac{1}{\beta} r(\mathbf{x}_1, y)\right)}{h_{\boldsymbol{\psi}_1}(\mathbf{x}_t, y, t)} - 1 \right) \boldsymbol{v}_t(\mathbf{x}_t \mid \mathbf{x}_1, y) \right\|_2^2 \right].$$

The guidance network for flow matching is more complex than that used in diffusion models. The estimation errors from two guidance networks may accumulate and ultimately degrade generation performance. To address this limitation, we propose a training-free guidance method for flow matching that mitigates these issues.

$$\boldsymbol{v}_t^q(\mathbf{x}_t, y)$$

$$= \boldsymbol{v}_t^p(\mathbf{x}_t, y) + \mathbb{E}_{\mathbf{x}_1 \sim p_{1|t}(\mathbf{x}_1 | \mathbf{x}_t, y)} \left[ \left( \frac{\exp\left(\frac{1}{\beta} r(\mathbf{x}_1, y)\right)}{\mathbb{E}_{\mathbf{x}_1' \sim p_{1|t}(\mathbf{x}_1' | \mathbf{x}_t, y)} \left[\exp\left(\frac{1}{\beta} r(\mathbf{x}_1', y)\right)\right]} - 1 \right) \boldsymbol{v}_t(\mathbf{x}_t \mid \mathbf{x}_1, y) \right]$$

$$= \boldsymbol{v}_t^p(\mathbf{x}_t, y) + \int_{\mathbf{x}_1} \left( \frac{\exp\left(\frac{1}{\beta} r(\mathbf{x}_1, y)\right)}{\mathbb{E}_{\mathbf{x}_1' \sim p_{1|t}} \left[\exp\left(\frac{1}{\beta} r(\mathbf{x}_1', y)\right)\right]} - 1 \right) \boldsymbol{v}_t(\mathbf{x}_t \mid \mathbf{x}_1, y) \, p_{1|t}(\mathbf{x}_1 \mid \mathbf{x}_t, y) \, d\mathbf{x}_1$$

$$= \boldsymbol{v}_t^p(\mathbf{x}_t, y) + \int_{\mathbf{x}_1} \left( \frac{\exp\left(\frac{1}{\beta} r(\mathbf{x}_1, y)\right)}{\mathbb{E}_{\mathbf{x}_1 \sim p_{1|t}} \left[\exp\left(\frac{1}{\beta} r(\mathbf{x}_1, y)\right)\right]} - 1 \right) \boldsymbol{v}_t(\mathbf{x}_t \mid \mathbf{x}_1, y) \, \frac{p_{t|1}(\mathbf{x}_t \mid \mathbf{x}_1, y) \, p(\mathbf{x}_1 \mid y)}{p_t(\mathbf{x}_t \mid y)} d\mathbf{x}_1$$

$$= \boldsymbol{v}_t^p(\mathbf{x}_t, y) + \mathbb{E}_{\mathbf{x}_1 \sim p(\mathbf{x}_1 | y)} \left[ \left( \frac{\exp\left(\frac{1}{\beta} r(\mathbf{x}_1, y)\right)}{\mathbb{E}_{\mathbf{x}_1 \sim p_{1|t}} \left[\exp\left(\frac{1}{\beta} r(\mathbf{x}_1, y)\right)\right]} - 1 \right) \boldsymbol{v}_t(\mathbf{x}_t \mid \mathbf{x}_1, y) \, \frac{p_{t|1}(\mathbf{x}_t \mid \mathbf{x}_1, y)}{p_t(\mathbf{x}_t \mid y)} \right]$$

$$= \boldsymbol{v}_t^p(\mathbf{x}_t, y) + \mathbb{E}_{\mathbf{x}_1 \sim p(\mathbf{x}_1 | y)} \left[ \left( \frac{\exp\left(\frac{1}{\beta} r(\mathbf{x}_1, y)\right)}{\mathbb{E}_{\mathbf{x}_1 \sim p_{1|t}} \left[\exp\left(\frac{1}{\beta} r(\mathbf{x}_0, y)\right)\right]} - 1 \right) \boldsymbol{v}_t(\mathbf{x}_t \mid \mathbf{x}_1, y) \, \frac{p_{t|1}(\mathbf{x}_t \mid \mathbf{x}_1, y)}{\mathbb{E}_{\mathbf{x}_1 \sim p(\mathbf{x}_1 | y)} \left[p_{t|1}(\mathbf{x}_t \mid \mathbf{x}_1, y)\right]} \right]$$

$$= \boldsymbol{v}_t^p(\mathbf{x}_t, y) + \mathbb{E}_{\mathbf{x}_1 \sim p(\mathbf{x}_1 | y)} \left[ \left( \frac{\exp\left(\frac{1}{\beta} r(\mathbf{x}_1, y)\right)}{\mathbb{E}_{\mathbf{x}_1 \sim p(\mathbf{x}_1 | y)} \left[\exp\left(\frac{1}{\beta} r(\mathbf{x}_1, y)\right) \frac{p_{t|1}(\mathbf{x}_t \mid \mathbf{x}_1, y)}{\mathbb{E}_{\mathbf{x}_1 \sim p(\mathbf{x}_1 | y)} \left[p_{t|1}(\mathbf{x}_t \mid \mathbf{x}_1, y)\right]}\right]} - 1 \right) \right.$$

$$\left. \boldsymbol{v}_t(\mathbf{x}_t \mid \mathbf{x}_1, y) \, \frac{p_{t|1}(\mathbf{x}_t \mid \mathbf{x}_1, y)}{\mathbb{E}_{\mathbf{x}_1 \sim p(\mathbf{x}_1 | y)} \left[p_{t|1}(\mathbf{x}_t \mid \mathbf{x}_1, y)\right]} \right].$$

$$\square$$

### A.3 PROOF OF LEMMA 3.2

*Proof.* The proof is straightforward and we include it below for completeness. Note that the objective function can be rewritten as

$$\mathcal{L}_{\text{guidance}}(\boldsymbol{\psi})$$

$$:= \mathbb{E}_{p(\mathbf{x}_0, \mathbf{x}_t, y)} \left[ \left\| h_{\boldsymbol{\psi}}(\mathbf{x}_t, y, t) - \exp\left(\frac{1}{\beta} r(\mathbf{x}_0, y)\right) \right\|_2^2 \right]$$

$$= \int_{\mathbf{x}_t} \int_y \left\{ \int_{\mathbf{x}_0} p(\mathbf{x}_0 | \mathbf{x}_t, y) \left\| h_{\boldsymbol{\psi}}(\mathbf{x}_t, y, t) - \exp\left(\frac{1}{\beta} r(\mathbf{x}_0, y)\right) \right\|_2^2 d\mathbf{x}_0 \right\} p(\mathbf{x}_t | y) p(y) dy d\mathbf{x}_t$$

$$= \int_{\mathbf{x}_t} \int_y \left\{ \|h_{\boldsymbol{\psi}}(\mathbf{x}_t, y, t)\|_2^2 - 2\langle h_{\boldsymbol{\psi}}(\mathbf{x}_t, y, t), \int_{\mathbf{x}_0} p(\mathbf{x}_0 | \mathbf{x}_t, y) \exp\left(\frac{1}{\beta} r(\mathbf{x}_0, y)\right) d\mathbf{x}_0 \rangle \right\} p(\mathbf{x}_t | y) p(y) dy d\mathbf{x}_t + C$$

$$= \int_{\mathbf{x}_t} \int_y \left\| h_{\boldsymbol{\psi}}(\mathbf{x}_t, y, t) - \mathbb{E}_{p(\mathbf{x}_0 | \mathbf{x}_t, y)} \left[\exp\left(\frac{1}{\beta} r(\mathbf{x}_0, y)\right)\right] \right\|_2^2 p(\mathbf{x}_t | y) p(y) dy d\mathbf{x}_t,$$

where $C$ is a constant independent of $\boldsymbol{\psi}$. Thus we have the minimizer $\boldsymbol{\psi}^* = \underset{\boldsymbol{\psi}}{\arg\min} \, \mathcal{L}_{\text{guidance}}(\boldsymbol{\psi})$

satisfies $h_{\boldsymbol{\psi}^*}(\mathbf{x}_t, y, t) = \mathbb{E}_{p(\mathbf{x}_0|\mathbf{x}_t, y)}\left[\exp\left(\frac{1}{\beta}r(\mathbf{x}_0, y)\right)\right]$. $\qquad\square$

## B   MORE DETAILS ON EXPERIMENTS

### B.1   ALGORITHMS FOR TRAINING THE GUIDANCE NETWORK

Algorithm 1 is the algorithm for training the guidance network.

---

**Algorithm 1** Algorithm for Training a Guidance Network

---

**Require:** Samples from alignment dataset, pre-trained one-step diffusion model $s(\mathbf{x}_T, y, T)$, pre-determined reward function $r(\mathbf{x}_0, y)$, hyperparameters $\eta, \beta$, and initial weights of guidance network $\boldsymbol{\psi}$.

1: **repeat**
2:     Sample mini-batch data from alignment dataset with batch size $b$.
3:     Perturb $\mathbf{x}_0$ using forward transition $p(\mathbf{x}_T|\mathbf{x}_0)$.
4:     Compute guidance loss:
5:

$$\mathcal{L}_{\text{guidance}}(\boldsymbol{\psi}) = \frac{1}{b}\sum_{\mathbf{x}_0, \mathbf{x}_T, y}\left\| h_{\boldsymbol{\psi}}(\mathbf{x}_T, y) - \exp\left(\frac{1}{\beta}r(\mathbf{x}_0, y)\right)\right\|_2^2.$$

6:     Sample mini-batch from winning responses $(\mathbf{x}', y)$ with batch size $b$.
7:     Perturb $\mathbf{x}_0'$ using forward transition $q(\mathbf{x}_T'|\mathbf{x}_0')$.
8:     Compute consistency loss:
9:

$$\mathcal{L}_{\text{consistence}} = \frac{1}{b}\sum_{\mathbf{x}_0', \mathbf{x}_T', y}\left\| s(\mathbf{x}_T', y, T) + \nabla_{\mathbf{x}_T'}\log h_{\boldsymbol{\psi}}(\mathbf{x}_T', y) - \nabla_{\mathbf{x}_T'}\log q(\mathbf{x}_T|\mathbf{x}_0', y)\right\|_2^2.$$

10:    Update $\boldsymbol{\psi}$ via gradient descent:

$$\nabla_{\boldsymbol{\psi}}\left(\mathcal{L}_{\text{guidance}} + \eta \, \mathcal{L}_{\text{consistence}}\right).$$

11: **until** convergence
12: **return** weights of guidance network $\boldsymbol{\psi}$.

---

### B.2   ABLATION STUDY ON HYPERPARAMETER

In this subsection, we provide the ablation study of the strength of the regularization $\eta$ and the strength of the reward function $\beta$ in the following table.

Table 3: Ablation study of hyperparameter on PickScore.

| $\eta$ | $\beta = 10$ | $\beta = 15$ | $\beta = 20$ |
|---|---|---|---|
| 0.1 | 22.82 | 22.79 | 22.72 |
| 0.5 | 22.78 | 23.01 | 22.79 |
| 1 | 22.76 | **23.08** | 22.84 |

Table 4: Prompts used to generate Figure 1.

| Image | Prompt |
|---|---|
| Col1 | Saturn rises on the horizon. |
| Col2 | a watercolor painting of a super cute kitten wearing a hat of flowers |
| Col3 | A galaxy-colored figurine floating over the sea at sunset, photorealistic. |
| Col4 | fireclaw machine mecha animal beast robot of horizon forbidden west horizon zero dawn bioluminiscence, behance hd by jesper ejsing, by rhads, makoto shinkai and lois van baarle, ilya kuvshinov, rossdraws global illumination |
| Col5 | A swirling, multicolored portal emerges from the depths of an ocean of coffee, with waves of the rich liquid gently rippling outward. The portal engulfs a coffee cup, which serves as a gateway to a fantastical dimension. The surrounding digital art landscape reflects the colors of the portal, creating an alluring scene of endless possibilities. |
| Col6 | A profile picture of an anime boy, half robot, brown hair |
| Col7 | Detailed Portrait of a cute woman vibrant pixie hair by Yanjun Cheng and Hsiao-Ron Cheng and Ilya Kuvshinov, medium close up, portrait photography, rim lighting, realistic eyes, photorealism pastel, illustration |
| Co18 | On the Mid-Autumn Festival, the bright full moon hangs in the night sky. A quaint pavilion is illuminated by dim lights, resembling a beautiful scenery in a painting. Camera type: close-up. Camera lens type: telephoto. Time of day: night. Style of lighting: bright. Film type: ancient style. HD. |

## B.3 PROMPTS FOR FIGURE IN MAIN PAPER

## B.4 NONE DIFFERENTIABLE REWARD

Since the proposed framework is model agnostic and reward agnostic. Our method can be applied to any one-step model and even a non-differentiable reward function. We adopt the GenEval dataset to further demonstrate the effectiveness of the proposed method. The GenEval dataset evaluates whether the generated images are aligned with the prompt regarding object co-occurrence, position, count, and color. We apply official GenEval scripts to generate 5k training prompts. We use SDXL-turbo to generate 10 images per prompt to construct the source dataset and select the correct text image pair as the target dataset for regularization. We train the guidance network for 10 epochs and get the results in Table 5. It verifies the general applicability of the proposed framework. Most importantly, the reward function of the GenEval dataset is binary (1 for correct, 0 for incorrect), which is not differentiable. The unbiased Monte Carlo estimation of the direct backpropagation method cannot be applied to this non-differentiable reward function.

Table 5: Performance on the GenEval benchmark. Our method consistently outperforms SDXL across all sub-tasks.

| Method | Single Obj. | Two Obj. | Counting | Colors | Position | Color Attr. | Overall |
|---|---|---|---|---|---|---|---|
| SDXL | 0.97 | 0.72 | 0.37 | 0.83 | 0.10 | 0.21 | 0.53 |
| Ours | **0.98** | **0.75** | **0.41** | **0.86** | **0.16** | **0.26** | **0.57** |

## LLM USAGE

LLMs are only used for polishing the writing. No ideas or discoveries are contributed by LLMs.

