# OpenReview forum: "Finetuning-free Alignment of Diffusion Model for Text-to-Image Generation"
_ICLR.cc/2026/Conference — Submitted to ICLR 2026_

### Official Review · Reviewer_z6Mz · 2025-10-30

**Soundness:** 2
**Presentation:** 2
**Contribution:** 2
**Rating:** 4
**Confidence:** 4

**Summary:**

This paper presents an approach to aligning text-to-image diffusion models with human preferences. Instead of relying on computationally expensive fine-tuning of the base model (such as DPO), the authors propose a lightweight, plug-and-play guidance mechanism. The key contributions include a diagnosis of why naive guidance methods often fail, attributed to the adversarial nature of the guidance signal, and a solution that trains a small, regularized guidance network to provide a stable, artifact-free signal for diffusion models.

**Strengths:**

- The finding that the adversarial nature of the guidance can lead to undesirable artifacts in the generated images is interesting.
- The ablation studies effectively demonstrate the effectiveness of each proposed component.

**Weaknesses:**

Despite the theoretical discussion in this work, there still lacks solid experiments to validate the proposed approach.

- The method's effectiveness has not been validated across different diffusion model architectures, leaving its generalizability to other frameworks unclear.

- The method's performance has not been demonstrated on other datasets, which limits claims of general applicability.

- The paper lacks a sensitivity analysis for its newly introduced hyperparameters, making the method's robustness to parameter variations unclear.

**Questions:**

Besides, there are formatting issues in Lines 72–74: the manuscript appears to contain white text, e.g., “ted in one or very few steps, the two samples would only exhibit small differences in details. SPM allows us to capture such detail differences and guide the diffusion model.”

This raises concerns about potential prompt injection targeting AI-assisted reviewers or, alternatively, author oversight in document preparation.

**Details Of Ethics Concerns:**

Please see #Questions about the white text in the manuscript.

---

> ### Author Response · Authors · 2025-11-25
>
> We thank the reviewer for the thoughtful comments and suggestions. We appreciate the time you spent on the paper and your recognition of our contributions and strengths. Below we address the concerns and comments that you have provided.
>
> **Q**: *The method's effectiveness has not been validated across different diffusion model architectures, leaving its generalizability to other frameworks unclear. *
>
> **A**:  Thank you very much for your question. We further adopt Stable Diffusion 3.5 large turbo to verify the effectiveness of the proposed method. The experimental results can be found in Figure 1and Table 2 in the revised version. Stable Diffusion 3.5 large turbo is one of the state-of-the-art models in text-to-image generation with a DiT-based model and flow matching. Our framework consistently improves the performance of SD3.5 large turbo on all criteria without finetuning, which demonstrates the generalization ability of the proposed framework.
>
> **Q**: *The method's performance has not been demonstrated on other datasets, which limits claims of general applicability.*
>
> **A**:  Thank you very much for your question. We would like to clarify that our experiments encompass a broader range of reward functions. We adopt Pickscore, HPS_V2, Imagereward, Aesthetic score, and non-differentiable GenEval. Additionally, we evaluate our method using prompts from both Pickscore and GenEval. This broad evaluation demonstrates that our approach generalizes across multiple reward functions and prompt sets, supporting its applicability beyond a single dataset.
>
> **Q**: *The paper lacks a sensitivity analysis for its newly introduced hyperparameters, making the method's robustness to parameter variations unclear.*
>
> **A**:  Thank you very much for your suggestions. We include the ablation study of hyperparameters $\beta$ and $\eta$ in Appendix B.2.
>
> **Q**: *Besides, there are formatting issues in Lines 72–74: the manuscript appears to contain white text, e.g., “ted in one or very few steps, the two samples would only exhibit small differences in details. SPM allows us to capture such detail differences and guide the diffusion model.”
> This raises concerns about potential prompt injection targeting AI-assisted reviewers or, alternatively, author oversight in document preparation.*
>
> **A**:  Thank you very much for your question. The authors confirm that there is no prompt injection in either the original or revised version. We are unsure how the reviewer extracted the quoted text (“ted in one or very few steps, the two samples would only exhibit small differences in details. SPM allows us to capture such detail differences and guide the diffusion model”) from our original manuscript. We have double-checked the formatting issue in lines 72–74, which is the caption of Figure 2 in SPO [1], since we directly adopt Figure 2 in SPO for comparison. We have corrected the formatting issue in the revised version.
>
> [1] Liang, Z., Yuan, Y., Gu, S., Chen, B., Hang, T., Cheng, M., Li, J., & Zheng, L. (2025). Aesthetic Post-Training Diffusion Models from Generic Preferences with Step-by-step Preference Optimization. CVPR.

---

### Official Review · Reviewer_9tVa · 2025-11-01

**Soundness:** 3
**Presentation:** 3
**Contribution:** 3
**Rating:** 8
**Confidence:** 3

**Summary:**

This paper proposes a finetuning-free alignment framework for text-to-image diffusion models that avoids the computational cost and limited generalization of existing RLHF- or DPO-based fine-tuning approaches. Instead of modifying model weights, the authors reinterpret preference alignment as sampling from a reward-weighted distribution, showing that the aligned score function can be decomposed into the original diffusion model score and an additional guidance term derived from a learned reward model. Experiments on text-to-image benchmarks demonstrate that the method achieves comparable or superior alignment quality to state-of-the-art fine-tuning methods.

**Strengths:**

S1. The paper tackles a principled decomposition of the aligned score function into the diffusion model score, combined with the reward-based guidance term. The proposed method provides a conceptually elegant connection between preference learning and inference-time guidance, clarifying the relationship between RLHF/DPO-style methods and diffusion sampling.

S2. The proposed method modifies neither the diffusion model parameters nor the text encoder, making it model-agnostic and straightforward to integrate with existing text-to-image pipelines.

S3. The paper’s method achieves strong alignment performance while avoiding the heavy training overhead required by RLHF or DPO approaches. Combined with Stable Diffusion XL-Turbo, the method supports one-step T2I generation, making the overall pipeline to be suitable for practical usage and user-interactive generation scenarios.


S4. The method consistently improves PickScore, HPS-v2, ImageReward, and Aesthetic score. Qualitative examples also show visually appealing improvements compared to the baselines.

**Weaknesses:**

W1. The proposed method assumes that the forward diffusion process remains unchanged when aligning the model to human preferences, meaning the aligned distribution $q(x_t | x_0)$ is assumed to match the original pretrained model’s noising process $p(x_t | x_0)$. This assumption effectively preserves the base diffusion model’s denoising trajectory, which determines the global layout, and object composition of the generated image. As a result, while the proposed method is well-suited for adjusting aesthetic properties or making small semantic refinements, it may struggle to generate plausible image output conditioned on prompts requiring strong semantic correction, multi-object reasoning, or compositional control (e.g., enforcing spatial relations or specific attribute assignments). I was wondering if the paper’s method can also handle such generation tasks.

W2. The guidance network outputs cannot be differentiated to denote where or which components of the image fail to match the textual specification. Consequently, the approach may struggle on prompts that involve explanations on multiple objects or spatial relations (*e.g.*, “to the left of,” “behind”).

W3. The authors use Stable Diffusion XL-Turbo for experiments. However, recent works use the Transformer-based diffusion model, beyond U-Net based Stable Diffusion XL. Is it possible to apply the proposed method into the DiT-based model, such as Stable Diffusion v3 or even FLUX?

W4. Because the diffusion backbone remains frozen, generated outputs closely reflect the inductive biases of the reward function. Is there any additional methods or strategy to alleviate the inductive biases of the given reward function, such as PickScore or Aesthetic Score?

**Questions:**

Please check the weakness section.

---

> ### Author Response · Authors · 2025-11-25
>
> We thank the reviewer for the thoughtful comments and suggestions. We appreciate the time you spent on the paper and your recognition of our contributions and strengths. Below we address the concerns and comments that you have provided.
>
> **Q**: *The proposed method assumes that the forward diffusion process remains unchanged when aligning the model to human preferences, meaning the aligned distribution $p(x_t|x_0)$ is assumed to match the original pretrained model’s noising process $p(x_t|x_0)$. This assumption effectively preserves the base diffusion model’s denoising trajectory, which determines the global layout, and object composition of the generated image. As a result, while the proposed method is well-suited for adjusting aesthetic properties or making small semantic refinements, it may struggle to generate plausible image output conditioned on prompts requiring strong semantic correction, multi-object reasoning, or compositional control (e.g., enforcing spatial relations or specific attribute assignments). I was wondering if the paper’s method can also handle such generation tasks.*
>
> **A**:  Thank you very much for your question. In general, assuming the aligned distribution $p(x_t|x_0)$ is the same as the original pretrained model’s noising process $p(x_t|x_0)$ is a very mild assumption and also adopted by [1,2,3,4]. It only requires
> that the base noise distribution $p_1(x_1) = \mathcal{N}(0, I)$ and the
> diffusion schedule remain unchanged. This choice does not constrain the model’s
> overall generative capacity. The key observation is that, even with an unchanged forward process, the data
> distribution is modified during alignment. As a result, the posterior
> $q_{0\mid t}(x_0 \mid x_t) \propto q(x_t \mid x_0)\, q_0(x_0)$
> changes in a meaningful way. Because the posterior depends directly on the
> aligned data distribution $q_0(x_0)$, it can encode substantial semantic
> adjustments. Consequently, the method can support strong semantic correction,
> multi-object reasoning, and compositional control whenever these preferences are
> reflected in the aligned data distribution.
>
>
> [1] Lu, C., Chen, H., Chen, J., Su, H., Li, C., & Zhu, J. (2023). Contrastive Energy Prediction for Exact Energy-Guided Diffusion Sampling in Offline Reinforcement Learning. ICML.
>
> [2] Song, J., Zhang, Q., Yin, H., Mardani, M., Liu, M., Kautz, J., Chen, Y., & Vahdat, A. (2023). Loss-Guided Diffusion Models for Plug-and-Play Controllable Generation. ICML.
>
> [3] Shen, Y., Jiang, X., Wang, Y., Yang, Y., Han, D., & Li, D. (2024). Understanding and Improving Training-free Loss-based Diffusion Guidance. NeurIPS.
>
> [4] Feng, R., Wu, T., Yu, C., Deng, W., & Hu, P. (2025). On the Guidance of Flow Matching. ICML.
>
> **Q**: *The guidance network outputs cannot be differentiated to denote where or which components of the image fail to match the textual specification. Consequently, the approach may struggle on prompts that involve explanations on multiple objects or spatial relations (e.g., “to the left of,” “behind”).*
>
> **A**:  Thank you very much for your question. In the preliminary experiments, we found that the performance of the guidance network also depends on the quality of the base model, especially for some complicated tasks like multiple objects or spatial relations (e.g., “to the left of,” “behind”) shown in Table 4. Although the reward functions are not differentiable, our proposed method still improves the performance on "Position" a little bit. We hypothesize that this is because the guidance network is trained on corrected image pairs (target distribution), which may help the pre-trained model better align with the intended spatial relationships.

---

> > ### Author Response · Authors · 2025-11-25
> >
> > **Q**: *The authors use Stable Diffusion XL-Turbo for experiments. However, recent works use the Transformer-based diffusion model, beyond U-Net based Stable Diffusion XL. Is it possible to apply the proposed method into the DiT-based model, such as Stable Diffusion v3 or even FLUX?*
> >
> > **A**:  Thank you very much for your suggestions. We adopt Stable Diffusion 3.5 large turbo to verify the effectiveness of the proposed method. The experimental results can be found in Figure 1and Table 2 in the revised version. Stable Diffusion 3.5 large turbo is one of the state-of-the-art models in text-to-image generation with a DiT-based model and flow matching. To adopt the guidance for SD3.5 large turbo, we provide the exact guidance in Theorem 3.3 in the revised version. Our framework consistently improves the performance of SD3.5 large turbo on all criteria without finetuning, which demonstrates the generalization ability of the proposed framework.
> >
> > **Q**: *Because the diffusion backbone remains frozen, generated outputs closely reflect the inductive biases of the reward function. Is there any additional methods or strategy to alleviate the inductive biases of the given reward function, such as PickScore or Aesthetic Score?*
> >
> > **A**:  Thank you very much for your question. In general, RHLF methods suffer from the inductive biases of the reward function. The post-trained model is more likely to overfit the reward function, which is a phenomenon commonly referred to as reward hacking. A standard strategy to mitigate this issue is to aggregate multiple reward models, which has been adopted by the recent works [1,2,3]. This strategy can be directly applied in our framework as well.
> >
> > Moreover, the authors would like to emphasize that our method keeps the backbone of the pre-trained diffusion model frozen, our model might be more robust to the inductive biases of the reward function. It can be reflected in Figure 1, the images generated by our method do not exhibit overly “AI-stylized” against post-training methods.

---

> > > ### Comment · Reviewer_9tVa · 2025-11-27
> > >
> > > Thanks for the detailed response. I have one question: the authors mentioned in the rebuttal that they included experimental results using Stable Diffusion 3.5 large Turbo in Figure 1 and Table 2 of the revised manuscript, but these updates do not seem to be reflected in the current PDF file. Additionally, the authors refer to “some complicated tasks like multiple objects or spatial relations shown in Table 4,” but I do not see relevant content in Table 4. It would be helpful if the authors could check these points, and if I am missing something, I would appreciate the clarification.

---

> > > > ### Author Response · Authors · 2025-11-27
> > > >
> > > > Thank you very much for your kind reminder. We just uploaded the revised version. If you have any questions, feel free to let us know.

---

### Official Review · Reviewer_r1pw · 2025-11-01

**Soundness:** 2
**Presentation:** 2
**Contribution:** 2
**Rating:** 4
**Confidence:** 3

**Summary:**

This paper proposes a finetuning-free method that improves the alignment of text-to-image diffusion models. It frames the alignment as a sampling problem from a reward-weighted distribution. Specifically, this paper decomposes the scoring function with a guidance term and proposes a regularization technique to train the model. Experimental results on Pick-a-Pic dataset show the improvement of the proposed method over baseline studies..

**Strengths:**

This paper proposes a finetuning-free method that is efficient compared to finetune-based methods. The proposed regularization strategy stabilizes the guidance signal and improves the  text-to-image diffusion models.

**Weaknesses:**

1. I have concerns regarding the evaluation of the proposed method.  According to line 418, the evaluation is conducted *using 500 validation prompts from the validation unique split of Pick-a-Pic.*  How are these prompts selected? Moreover, the baseline method SPO is evaluated on 4K prompts from Pick-a-Pic, which is eight times more than this method.

2. This paper evaluates its method based on SDXL-Turbo, which was released in 2023. Considering the rapid emergence of new models,  SDXL-Turbo is kind of 'old' and cannot well support the effectiveness and generalization of the proposed method. How does the proposed method perform when generalized to recent models?

3. Figure 1 shows some visualization results, while the prompts are provided in the appendix. It is kind of difficult for me to find the improvement of the proposed method over baselines. It seems the baseline method already gets good enough results.

4. Is it expected to include the related work in Section 1.1 instead of Section 2?

5. The citations of the paper could be improved, such as line 107
>  In (Liang et al., 2024), Liang et al. propose....

**Questions:**

Please refer to the weaknesses.

---

> ### Author Response · Authors · 2025-11-25
>
> We thank the reviewer for the thoughtful comments and suggestions. We appreciate the time you spent on the paper and your recognition of our contributions and strengths. Below we address the concerns and comments that you have provided.
>
> **Q**: *I have concerns regarding the evaluation of the proposed method. According to line 418, the evaluation is conducted using 500 validation prompts from the validation unique split of Pick-a-Pic. How are these prompts selected? Moreover, the baseline method SPO is evaluated on 4K prompts from Pick-a-Pic, which is eight times more than this method.*
>
> **A**:  Thank you very much for your question. We utilize the same evaluation pipeline as SPO. SPO utilizes 500 validation prompts, i.e., validation unique split of Pick-a-Pic (which can be found in the second paragraph on page 6 of SPO).
>
> **Q**: *This paper evaluates its method based on SDXL-Turbo, which was released in 2023. Considering the rapid emergence of new models, SDXL-Turbo is kind of 'old' and cannot well support the effectiveness and generalization of the proposed method. How does the proposed method perform when generalized to recent models?*
>
> **A**:  Thank you very much for your suggestions. We adopt Stable Diffusion 3.5 large turbo to verify the effectiveness of the proposed method. The experimental results can be found in Figure 1 and Table 2 in the revised version. Stable Diffusion 3.5 large turbo is one of the state-of-the-art models in text-to-image generation with a DiT-based model and flow matching. To adopt the guidance for SD3.5 large turbo, we provide the exact guidance in Theorem 3.3 in the revised version. Our framework consistently improves the performance of SD3.5 large turbo on all criteria without finetuning, which demonstrates the generalization ability of the proposed framework.
>
> **Q**: *Figure 1 shows some visualization results, while the prompts are provided in the appendix. It is kind of difficult for me to find the improvement of the proposed method over baselines. It seems the baseline method already gets good enough results.*
>
> **A**:  Thank you very much for your question. In general, modern text-to-image diffusion models already produce high-quality images. However, we would like to highlight two aspects where our method offers improvements over existing baselines.
>
> 1. The first aspect is the alignment of the generated images with the prompt. For example, for the second column in Figure 1, the text prompt is "a watercolor painting of a super cute kitten wearing a hat of flowers", which means we expect the flowers to appear in the hat. The existing methods cannot deal with this appropriately, whereas our method produces a hat that contains noticeably more flower details, more faithfully reflecting the semantics of the prompt.
>
> 2. Many reward-based alignment approaches risk overfitting to the reward model (i.e., reward hacking), which often leads to overly “AI-stylized” or unnatural-looking images. Because our method keeps the diffusion backbone completely frozen, it retains the base model’s natural visual priors. As shown in Figure 1, the images produced by our method maintain a realistic style and do not exhibit the exaggerated AI-generated characteristics sometimes seen in post-trained baselines.
>
> **Q**: *Is it expected to include the related work in Section 1.1 instead of Section 2?*
>
> **A**:  Thank you very much for your suggestions. We would like to include the related work as a subsection of the Introduction, which can also be included in a new Section 2 if preferred.
>
> **Q**: *The citations of the paper could be improved, such as line 107*
>
> **A**:  Thank you very much for your suggestions. We improved the citations of the paper (lines 102, 105, 112, and 179) in the revised version.

---

### Official Review · Reviewer_ZRAs · 2025-11-02

**Soundness:** 2
**Presentation:** 2
**Contribution:** 2
**Rating:** 4
**Confidence:** 3

**Summary:**

This paper introduces a finetuning-free, plug-and-play alignment strategy for diffusion models in text-to-image generation by casting the problem as sampling from a reward-weighted distribution. The authors analyze the challenges of existing guidance-based alignment schemes—particularly the emergence of adversarial artifacts—and propose a novel regularization for guidance signal stabilization. The method is evaluated on established text-to-image benchmarks, achieves strong alignment to human preferences using a lightweight guidance network, and demonstrates substantial computational savings over finetuning-based approaches.

**Strengths:**

- **Formulation Innovation:** The paper reframes text-to-image alignment as direct sampling from a reward-weighted distribution, moving away from common parameter fine-tuning approaches and offering a generic plug-and-play control mechanism.
- **Practical estimator for the guidance:** The paper adopts a simple regression trick (Eq. 13) to approximate the conditional expectation and then converts it to a guidance gradient (Eq. 14), avoiding expensive backprop through the sampler.
- **Stabilization for adversarial guidance:** The instability of naïve guidance with increasing strength is documented and addressed via a consistency regularizer merged in Eq. 16.
- **Lightweight and fast:** The guidance net is only ~72 MB and reuses the reference model's VAE/tokenizer/text encoder. Combined with SDXL-Turbo, this enables effective **one-step** generation.
- **Agnostic to Reward:** The method supports both differentiable and non-differentiable reward settings, with Table 4 in the appendix demonstrating applicability on GenEval with binary rewards.

**Weaknesses:**

1. **Analysis Depth of Regularization:** While the regularization is empirically justified and its effect visualized (see Figure 2), the theoretical underpinnings and limits of this regularization are not fully elucidated. What modes of artifact are suppressed, and does the regularization always guarantee avoidance of adversarial guidance? The practical selection of the regularization hyperparameter $\eta$ (Eq. 13 & 15) also remains ad hoc.
2. **Reward Dependence & Generality:** Although the proposed scheme is reward-agnostic in form, its empirical evaluation—especially in Table 2—is predominantly based on PickScore and similar human-preference proxies. It is unclear how robust the approach is to poorly calibrated, biased, or low-signal rewards. There is only a narrow demonstration on non-differentiable rewards in Table 4 (GenEval), which is limited in scope and size.
3. **Scope of generalization is narrow:** Experiments are concentrated on SDXL-Turbo; the paper asserts model-agnosticism and one-step benefits, but offers limited cross-backbone verification or stress tests on distribution shift.
4. **Hyperparameter Sensitivity:** The proposed method claims to "fix" the problem of carefully tuning the guidance strength, but practical recipes or robustness studies for the guidance parameter, regularization weight, or hyperparameter $\beta$ are lacking.
5. **Comparisons to very recent other alignment methods are light:** Table 2 includes Tweedie/Backprop and two finetuning methods (Diffusion-DPO, SPO), but a broader slate of strong alignment related methods (and best-practice configs) would better establish relative advantage.
6. **Not Strictly Finetuning-free:** Please refer to the precise definition of finetuning-free. The scenario described in this paper can at best be considered "no base-model fine-tuning".

**Questions:**

1. **Regularization Mechanics:** Can the authors provide more intuition on how the proposed regularization term shapes the guidance network’s landscape? Are there scenarios or reward functions where this regularization might fail or even worsen adversarial behaviors?
2. **Sensitivity Analysis:** How does performance vary with η and β? Please provide curves (PickScore/HPSV2/ImageReward/Aesthetic vs. η, β) and report variance across seeds.
3. **Extension to Other Backbones:** Have you tested non-Turbo SDXL or SD 2.1 latent backbones, or text-conditional DiT variants(like Flux)? Are there empirical results or qualitative observations on data distribution shifts not covered by the current benchmarks?
4. **Robustness to Reward Misspecification:** Beyond GenEval's binary reward, how does the method fare under noisy, sparse, or biased rewards? Can the guidance network overfit to reward artifacts, and how would the regularizer respond?
5. **Comparisons with Other Alignment Related Work:** Where are the practical/theoretical boundaries vs. other plug-and-play or inference-time guidance alignment related methods? This will help substantiate the method's effectiveness and superiority.

---

> ### Author Response · Authors · 2025-11-25
>
> We thank the reviewer for the thoughtful comments and suggestions. We appreciate the time you spent on the paper and your recognition of our contributions and strengths, including the motivation, solid theoretical analysis, clear presentation, and effective ablation studies. Below we address the concerns and comments that you have provided.
>
> **Q**: *Analysis Depth of Regularization: While the regularization is empirically justified and its effect visualized (see Figure 2), the theoretical underpinnings and limits of this regularization are not fully elucidated. What modes of artifact are suppressed, and does the regularization always guarantee avoidance of adversarial guidance? The practical selection of the regularization hyperparameter $\eta$ (Eq. 13 & 15) also remains ad hoc. Sensitivity Analysis: How does performance vary with η and β? *
>
> **A**:  Thank you very much for your questions. The regularization term in our framework is score matching on the target distribution. Therefore, it admits the same optimal solution as the training objective of the guidance network. This theoretical equivalence ensures that the regularization does not change the optimal solution. In practice, the optimal choice of η depends on the relative sample sizes from the source and target distributions, as this affects the estimation variance of both terms.
>
> **Q**: *Reward Dependence & Generality: Although the proposed scheme is reward-agnostic in form, its empirical evaluation—especially in Table 2—is predominantly based on PickScore and similar human-preference proxies. It is unclear how robust the approach is to poorly calibrated, biased, or low-signal rewards. There is only a narrow demonstration on non-differentiable rewards in Table 4 (GenEval), which is limited in scope and size.*
>
> **A**:  Thank you very much for your questions. We would like to clarify that our evaluation encompasses a broader range of reward functions. We adopt pickscore, HPS_V2, Imagereward, Aesthetic score, and non-differentiable GenEval. This broad evaluation demonstrates that our approach generalizes across multiple reward functions and prompt sets.
>
> **Q**: *Scope of generalization is narrow: Experiments are concentrated on SDXL-Turbo; the paper asserts model-agnosticism and one-step benefits, but offers limited cross-backbone verification or stress tests on distribution shift. Extension to Other Backbones: Have you tested non-Turbo SDXL or SD 2.1 latent backbones, or text-conditional DiT variants(like Flux)? Are there empirical results or qualitative observations on data distribution shifts not covered by the current benchmarks?*
>
> **A**:  Thank you very much for your suggestions. We adopt Stable Diffusion 3.5 large turbo to verify the effectiveness of the proposed method. The experimental results can be found in Figure 1 and Table 2 in the revised version. Stable Diffusion 3.5 large turbo is one of the state-of-the-art models in text-to-image generation with a DiT-based model and flow matching. To adopt the guidance for SD3.5 large turbo, we provide the exact guidance in Theorem 3.3 in the revised version. Our framework consistently improves the performance of SD3.5 large turbo on all criteria without finetuning, which demonstrates the generalization ability of the proposed framework.

---

> > ### Author Response · Authors · 2025-11-25
> >
> > **Q**: *Hyperparameter Sensitivity: The proposed method claims to "fix" the problem of carefully tuning the guidance strength, but practical recipes or robustness studies for the guidance parameter, regularization weight, or hyperparameter $\beta$ are lacking. The practical selection of the regularization hyperparameter $\eta$ (Eq. 13 & 15) also remains ad hoc. Please provide curves (PickScore/HPSV2/ImageReward/Aesthetic vs. η, β) and report variance across seeds.*
> >
> > **A**:  Thank you very much for your suggestions. We include the ablation study of hyperparameters $\beta$ and $\eta$ in Appendix B.2.
> >
> > **Q**: *Comparisons to very recent other alignment methods are light: Table 2 includes Tweedie/Backprop and two finetuning methods (Diffusion-DPO, SPO), but a broader slate of strong alignment related methods (and best-practice configs) would better establish relative advantage. Comparisons with Other Alignment Related Work: Where are the practical/theoretical boundaries vs. other plug-and-play or inference-time guidance alignment related methods? This will help substantiate the method's effectiveness and superiority.*
> >
> > **A**: Thank you very much for your questions. We believe that including both Tweedie’s formula and Direct Backpropagation together with two finetuning methods(Diffusion-DPO, SPO) in Tables 1 and 2 is an important strength of our paper. We conducted a fair and controlled comparison by evaluating all methods on the same backbone, some prompts, and the same reward function. In addition, we carefully tuned the hyperparameters for Tweedie’s formula and Direct Backpropagation to avoid the undesirable artifacts illustrated in Figure 2. The proposed method is unbiased with a step size guarantee and also achieves great performance on PickScore, HPSV2, ImageReward, and Aesthetic with a 60\% reduction in computational cost.
> >
> > **Q**: *Not Strictly Finetuning-free: Please refer to the precise definition of finetuning-free. The scenario described in this paper can at best be considered "no base-model fine-tuning".*
> >
> > **A**:  Thank you very much for your suggestions. We emphasize the no base-model fine-tuning in line 379 in the revised version.

---

### Meta-Review · Area_Chair_nYHf · 2026-01-07

**Summary:**

This paper presents a framework for finetuning-free alignment for text-to-image diffusion models.
The submission received mixed but slightly negative reviews.
The reviewers mainly recognize the new formulation to reframe alignment, efficiency and speed with one-step generation, and good empirical results.
The main concerns from the reviewers were the limited generalization to newer backbones e.g. Flux or SD3 (all), sensitivity to hyperparameters (ZRAs and z6Mz), limited variety in evaluation data (r1pw and z6Mz), unclear justification of regularization (ZRAs), and biases from the underlying reward functions (ZRAs and 9tVa).
After reading the paper, the reviewers' comments and the authors' rebuttal, the AC believes the authors' responses would have partially addressed the reviewers' concerns, but there would still be outstanding concerns regarding limited evaluation scale and diversity and robustness to biased or noisy rewards. The theoretical grounding of regularization also remains ad hoc. The AC believes the merits do not significantly outweigh the weaknesses that warrants a clear acceptance at this time.

**Reviewer Concerns:**

Reviewers' concerns mostly addressed:
- Model generalization (all)
- Hyperparameter sensitivity (ZRAs, z6Mz)

Partially addressed concerns:
- Evaluation scale (r1pw)
- Comparison against recent methods (ZRAs)

Outstanding concerns:
- Justification of regularization & bias (ZRAs)
- Reward dependence & bias (ZRAs, 9tVa)
- Dataset diversity (z6Mz)

**Reviewer Scores:**

I think all reviewers would keep their original ratings.

---

### Decision · Program_Chairs · 2026-01-26

Reject